# Large Language Models as End-to-end Combinatorial Optimization Solvers

**Xia Jiang**
Eindhoven University of Technology
`x.jiang1@tue.nl`

**Yaoxin Wu**[*]
Eindhoven University of Technology
`y.wu2@tue.nl`

**Minshuo Li**
Eindhoven University of Technology
`m.li7@tue.nl`

**Zhiguang Cao**
Singapore Management University
`zgcao@smu.edu.sg`

**Yingqian Zhang**
Eindhoven University of Technology
`YQZhang@tue.nl`

## Abstract

Combinatorial optimization (CO) problems, central to decision-making scenarios like logistics and manufacturing, are traditionally solved using problem-specific algorithms requiring significant domain expertise. While large language models (LLMs) have shown promise in automating CO problem solving, existing approaches rely on intermediate steps such as code generation or solver invocation, limiting their generality and accessibility. This paper introduces a novel framework that empowers LLMs to serve as end-to-end CO solvers by directly mapping natural language problem descriptions to solutions. We propose a two-stage training strategy: supervised fine-tuning (SFT) imparts LLMs with solution generation patterns from domain-specific solvers, while a feasibility-and-optimality-aware reinforcement learning (FOARL) process explicitly mitigates constraint violations and refines solution quality. Evaluation across seven NP-hard CO problems shows that our method achieves a high feasibility rate and reduces the average optimality gap to 1.03–8.20% by tuning a 7B-parameter LLM, surpassing both general-purpose LLMs (e.g., GPT-4o), reasoning models (e.g., DeepSeek-R1), and domain-specific heuristics. Our method establishes a unified language-based pipeline for CO without extensive code execution or manual architectural adjustments for different problems, offering a general and language-driven alternative to traditional solver design while maintaining relative feasibility guarantees.

## 1 Introduction

Large language models (LLMs) have emerged as powerful tools with the potential to revolutionize not only traditional natural language processing (NLP) problems, but also a broad range of decision-making tasks, such as time series prediction [1, 2], medical diagnostics [3, 4], and computational optimization [5, 6]. As LLMs are increasingly recognized as general-purpose assistants for decision-making, there is growing interest in their potential to tackle NP-hard combinatorial optimization (CO) problems, which are central to many common real-world application scenarios. Specifically, CO arises in various domains such as transportation [7], manufacturing [8], and healthcare [9], often

---

[*]Corresponding author

requiring the design of complex heuristics to balance optimality and computational efficiency. LLMs facilitate CO by automating the problem-solving process, thereby reducing reliance on domain-specific modeling experts [5]. Ideally, an LLM can interpret user requests and generate corresponding solutions entirely through natural language. Thus, it reduces or eliminates the need for coding or formal modeling and making CO accessible to scenarios involving non-experts.

Existing research primarily leverages LLMs to solve CO problems by generating executable code to discover heuristics [6, 10, 11] or by interacting with optimization solvers [12, 13]. However, these methods still require substantial domain-specific expertise, as users must define problem-specific algorithmic templates (to evolve and discover heuristics) or proficiently use the existing optimizers (to execute the generated solver-calling programs). In contrast, end-to-end solvers model CO using natural language and offer a more appealing alternative by directly generating solutions, thus mitigating reliance on domain knowledge and offering a unified solving process [14].

While LLMs can reason out simple mathematical problems [15, 16], their end-to-end capability for CO remains highly limited [17, 18]. NP-hard CO problems often involve multiple constraints, making it challenging for LLMs to reason and identify even feasible solutions—let alone generate optimal ones. For example, GPT-3.5-turbo achieves an average optimality gap of 133% on the traveling salesman problem (TSP) with 50 nodes [19]. Even state-of-the-art models, such as GPT-4o and Claude-3.5-Sonnet, achieve only around a 50% feasibility rate on CO problems with fewer than 30 nodes—a limitation that persists even with the application of instruction tuning [20]. These persistent shortcomings in both optimality and feasibility have raised significant concerns regarding the potential of LLMs for solving CO problems [21]. Thus, it remains an open question on how to enable LLMs to effectively map natural language-described CO problems to high-quality solutions.

In this paper, we aim to empower LLMs to serve as end-to-end solvers for general CO problems via a two-stage fine-tuning approach. In the first stage, we use the supervised fine-tuning (SFT) paradigm to teach LLMs solution generation patterns by learning from domain-specific CO solvers. Our empirical results reveal that SFT alone leads to over-greedy behavior for some problems, where LLMs violate constraints in pursuit of improved objectives. To address this issue, we propose a feasibility-and-optimality-aware reinforcement learning (FOARL) method in the second stage, explicitly mitigating constraint violations while promoting better optimization outcomes. By training only lightweight Low-Rank Adaptation (LoRA) modules, our approach enables a 7B-parameter LLM to not only outperform the most advanced reasoning models, such as Deepseek-R1 [22] and OpenAI GPT-o1, but also surpass commonly used heuristics across seven CO problems.

Our contributions are outlined as follows. 1) Unlike recent approaches that rely on intermediate steps such as heuristic guidance or solver invocation, we model CO as a language generation task and develop an SFT strategy that fine-tunes LLMs to function as end-to-end solvers. This reduces dependence on domain-specific knowledge and simplifies the problem-solving pipeline. 2) Given that the SFT alone leads to over-greedy policies that violate problem-specific constraints, we introduce the FOARL algorithm, which refines the LLM policy using reinforcement learning (RL) to improve solution quality. 3) We evaluate the effectiveness of our method on seven different problems without manual architectural customization for the policy model, making a significant step towards creating a unified solver for CO based on language generation. Experimental results show that our fine-tuned 7B-parameter LLM can outperform various larger LLMs and problem-specific algorithms.

## 2 Related Work

**Combinatorial Optimization.** CO problems represent an important branch of optimization problems that aims to find the optimal solution from a finite set of candidates. Due to the NP-hard nature, exact solutions to CO are often computationally intractable, prompting the development of approximation and heuristic algorithms [23]. Given varying characteristics (e.g., objectives and constraints) of different CO problems, heuristic designs are typically problem-specific, such as the insertion algorithm for the TSP [24] and the savings algorithm for the vehicle routing problem (VRP) [25]. However, the design of these algorithms is typically a labor-intensive process that requires deep domain expertise [26]. Thus, there has been growing interest in automating the heuristic design process to reduce dependence on domain-specific knowledge. Recent advances have explored leveraging neural networks for CO, known as neural combinatorial optimization (NCO), which has shown promising performance on various problems [27–31]. In contrast, LLMs offer a more

user-friendly, natural language-based approach to interactive problem-solving, benefiting from the extensive knowledge learned during pre-training, thus opening up new opportunities for CO [14].

**LLMs for Optimization.** LLMs can function as either program generators or end-to-end optimizers for CO. Recent advances demonstrate that LLMs are capable of generating Python programs to iteratively discover heuristics or refine metaheuristics through evolutionary operations [6, 10, 32]. In addition, the LLM-based model-then-solve methods produce executable code that interfaces with CO solvers, such as Gurobi and OR-Tools, to solve problems described in natural language [5, 12, 33]. However, these approaches typically depend on complex prompt engineering and require users to be proficient in running and debugging the generated code. In contrast, treating LLMs as end-to-end optimization solvers offers a more automated and user-friendly alternative, which aims to directly map language-based inputs to solutions by: 1) zero-shot or few-shot generation [18, 20], 2) iterative refinement of initial solutions [19, 34], or 3) step-by-step solution construction [35]. Nonetheless, these methods are generally limited to small-scale CO problems (e.g., TSP with fewer than 30 nodes) and suffer from significant performance degradation as problem size increases [19]. Moreover, their target CO problems often lack complex constraints, such as the capacity constraint in VRP, which LLMs can easily violate due to hallucination. These limitations highlight the challenges in scaling end-to-end LLM solvers to more realistic and constrained optimization settings.

**LLMs for Mathematical Reasoning.** More generally, mathematical problems (e.g., those in algebra and optimization tasks) are essential benchmarks for evaluating LLMs' reasoning capabilities [36]. LLMs have been evaluated on math-related tasks, ranging from middle school-level arithmetic [37], Olympiad-level problems [38], and mathematical proofs [39], to combinatorial and graph optimization challenges [20, 35, 40]. Recent advances in LLM-based mathematical reasoning have been driven by structured prompting techniques, such as chain of thought (CoT) [41], tree of thought [42], and graph of thought [43]. These methods emphasize stepwise reasoning, in which natural language and logical operations are intertwined to explicitly generate interpretable chains of thought [16]. However, step-by-step reasoning becomes impractical for CO problems, particularly large-scale ones. Due to their NP-hard nature, maintaining feasible and high-quality solutions throughout a long reasoning trajectory is extremely challenging, resulting in substantial complexity and inefficiency when scaling to larger instances. In this paper, we propose an alternative paradigm that enables LLMs to directly learn end-to-end solution generation by imitating domain-specific solvers. Additionally, we incorporate heuristic information into the prompt to facilitate latent solution-space exploration, allowing LLMs to generate effective solutions without relying on explicit and verbose reasoning.

## 3 Problem Statement

We investigate how to apply LLMs to solve CO problems without relying on traditional algorithm design. Let $\mathcal{P}$ represent a specific CO problem (e.g., the TSP), with each instance $p \in \mathcal{P}$ defined as a tuple $(X_p, f_p, C_p)$, where $X_p$ is the set of all possible solutions (i.e., solution space); $f_p : X_p \to \mathbb{R}$ denotes the objective function to be minimized (or maximized); $C_p = \{c_{p,1}, c_{p,2}, \ldots, c_{p,m_p}\}$ is a set of constraints, and each $c_{p,i} : X_p \to \{0, 1\}$ indicates whether a solution satisfies the corresponding constraint or not. A feasible solution to a CO problem must satisfy all the involved constraints. For a given instance $p$, the goal is to find a solution $x_p^* \in X_p$ such that 1) $x_p^*$ satisfies all constraints in $C_p$, i.e., $\forall c_{p,i} \in C_p, c_{p,i}(x_p^*) = 1$, 2) for minimization problems, $f_p(x_p^*) \leq f_p(x)$ holds for all $x \in X_{C_p}$, where $X_{C_p} = \{x \in X_p | \forall c_{p,i} \in C_p, c_{p,i}(x) = 1\}$ is the set of all feasible solutions.

Let $\pi_\theta : T \to T$ be an LLM, where $T$ is the space of text sequences. We define two mapping functions: $\phi : \mathcal{P} \to T$, which maps a CO problem instance to its textual description, and $\psi_p : T \to X_p$, which maps text to a solution in the original solution space for $p$ (In practice, $\psi_p$ can be implemented using regular expressions or other parsing methods to extract solutions from the generated text). The end-to-end solution for a natural language-described CO problem is formulated as:

$$\hat{x}_p = \psi_p(\pi_\theta(\phi(p))), \tag{1}$$

where $\hat{x}_p$ is the LLM-generated solution to problem instance $p$. We evaluate the effectiveness of the end-to-end solution generation on a set of instances $\mathcal{P}_s$ using the average optimality gap $M_o(\mathcal{P}_s) = \frac{1}{|\mathcal{P}_s|} \sum_{p \in \mathcal{P}_s} \frac{f_p(\hat{x}_p) - f_p(x_p^*)}{|f_p(x_p^*)|}$ and feasibility rate $M_f(\mathcal{P}_s) = \frac{|\{p \in \mathcal{P}_s | \forall c_{p,i} \in C_p, c_{p,i}(\hat{x}_p) = 1\}|}{|\mathcal{P}_s|}$.

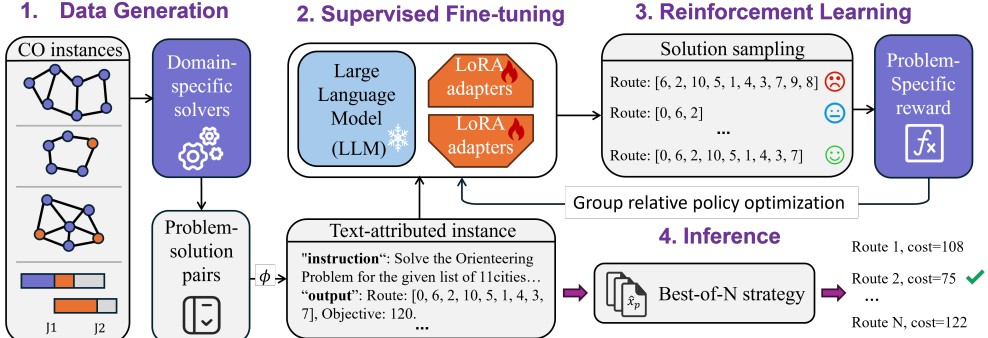

Figure 1: The framework of the proposed method.

In this paper, we consider CO problems, including TSP, orienteering problem (OP), capacitated VRP (CVRP), maximal independent set (MIS), minimum vertex cover (MVC), permutation flow shop scheduling problem (PFSP), and job-shop scheduling problem (JSSP). Notably, unlike traditional algorithm designs, which typically require customization in problem-specific heuristics or neural network architectures [44], our approach is generalizable and capable of solving other arbitrary (natural language-described) CO problems without extensive manual crafting or customization.

## 4 Methodology

### 4.1 Data Generation

The framework of the proposed method is illustrated in Figure 1. LLMs need to learn solution generation patterns from problem solutions, so the first step is generating high-quality data, i.e., problem-solution pairs. To this end, the solutions to different CO problems are produced by specialized solvers, e.g., the Lin-Kernighan-Helsgaun (LKH) solver [45] for TSP and the Q-learning-based iterated greedy (QIG) method [46] for PFSP—each incorporating distinct internal programmatic logic. Since it is intractable to unify the heterogeneous solving processes of these solvers by learning their byzantine internal logic, we use the solvers to generate (near)optimal solutions and train the LLM to directly learn from the solution representations, bypassing the need to replicate solver internals.

We generate numerical instances and solve them using domain-specific CO solvers. Each numerical instance is then transformed into natural language by a mapping function $\phi$, which is adapted from the text-attributed instance (TAI) framework in [14]. A TAI consists of two components: (1) Instruction: a problem description that outlines the optimization objective and relevant constraints, and (2) Input: instance-specific details that describe the attributes of individual nodes, for example, the city coordinates in a TSP instance. The outputs generated by the solvers, which are taken as the labels for SFT, are also converted into natural language by listing the solution and its corresponding objective. For example, the textual solution for a TSP instance is formatted as: "Routes: [0, 3, 5, 4, 1, 2, 6, 0], Objective: 5797.33". Instead of explicitly specifying the solution exploration through CoT, we facilitate latent solution-space exploration by providing the general heuristic features in the input prompt, which biases the model toward more promising regions of CO problems. Taking TSP as an example, this includes the top-$k$ ($k = 2$) neighbors for each node along with their respective distances. During training, the LLMs can learn to explore the latent space through informative and low-cost guidance of the heuristic features. Specifically, the problem definition, instance generation, utilized solver, and the example of TAI for the studied problems are provided in Appendix A.

### 4.2 Supervised Fine-tuning

In most NLP domains, fine-tuning LLMs relies on human-annotated data or detailed CoT reasoning trajectories, which are often costly and difficult to obtain [47]. Our SFT process for CO benefits from a more convenient data generation pipeline by leveraging existing specialized solvers. In this regard, we collect the datasets of problem-solution pairs without much manual effort, while enabling LLMs to effectively learn high-quality solution generation. We adopt the instruction tuning paradigm, which

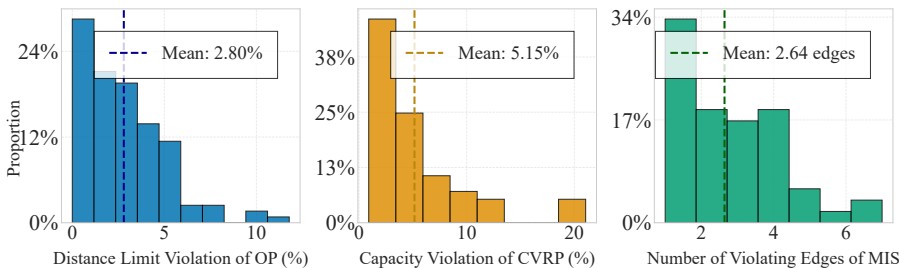

Figure 2: The extent of constraint violation of the SFT policy for different problems.

has proven effective in the optimization domain [13, 48], to equip LLMs with the ability to follow problem instructions and generate natural language descriptions of solutions provided by solvers.

Specifically, we formulate CO as a next-token prediction task, treating it as a language generation problem where the LLM learns to map textual problem descriptions to textual solution representations. For each CO problem instance $p$, we input the prompt $\phi(p)$ (i.e., the TAI described in Section 4.1). The target output is the natural language description of the solution $\psi_p^{-1}(x_p^*)$, where $x_p^*$ is the high-quality solution generated by a solver. Let $\theta$ be the trainable parameters of the LLM policy, and we minimize the standard language modelling loss during training:

$$\mathcal{L}_{\text{SFT}}(\theta) = -\sum_{i=1}^{n} \log \text{Pr}_\theta(y_i|\phi(p), y_{<i}), \tag{2}$$

where $y_i$ is the $i$-th token in the textual solution, and $y_{<i}$ denotes all previous tokens of the solution.

Our training data encompasses a range of problem sizes and distribution settings, promoting generalization across diverse CO problem instances. To ensure computational efficiency while preserving model performance, we employ LoRA [49], which introduces trainable matrices into the LLM while keeping most of the pre-trained weights frozen. The details of LoRA are provided in Appendix B.

However, SFT alone sometimes leads to over-greedy behavior, where the LLM learns toward marginally violating specific constraints in pursuit of improved objective values. We collect infeasible solutions resulting from constraint violations across 300 instances of OP, CVRP, and MIS, which are representative CO problems to illustrate this issue. As shown in Figure 2, a comparison between LLM-generated infeasible solutions and optimal ones reveals that most violations stem from slight oversteps of constraints, for example, traveling marginally farther in OP or serving slightly more customer demand in CVRP. These constraint violations are a primary cause of solution infeasibility in practice. This limitation motivates our subsequent reinforcement learning approach in Section 4.3.

### 4.3 Feasibility-and-optimality-aware Reinforcement Learning

Since LLMs trained via SFT primarily learn to generate natural language descriptions of solutions without explicit information about problem constraints, they often lack a comprehensive understanding of solution feasibility. Thus, the outputs may be infeasible or suboptimal, particularly in constrained CO tasks. To address this issue, we propose the FOARL algorithm to mitigate the greedy bias introduced by SFT. It enhances the LLM's reliability as the end-to-end CO solver by fostering feasibility awareness and correcting infeasible behaviors through the exploratory "trial-and-error" nature of RL. More precisely, the state of the RL corresponds to $\phi(p)$ for each instance $p$, while the action is the token predictions during generation. We define a feasibility-aware reward $\mathcal{R}_f^\mathcal{P}$ based on the constraint set $C_p = \{c_{p,1}, c_{p,2}, \ldots, c_{p,m_p}\}$, encouraging the model to recognize and satisfy all $m_p$ problem-specific constraints. To ensure that the model also maintains optimality, we introduce an optimality-aware reward $\mathcal{R}_o^\mathcal{P}$. These two rewards are jointly used to balance feasibility and optimality.

Specifically, we define the feasibility-aware reward function for any LLM-generated solution $\hat{x}_p$ as:

$$\mathcal{R}_f^\mathcal{P}(\hat{x}_p) = \begin{cases} \omega_0 \zeta + \sum_{i=1}^{m_p} \omega_i c_{p,i}(\hat{x}_p) & \text{if } \zeta \neq 0 \\ 0 & \text{if } \zeta = 0 \end{cases}, \forall p \in \mathcal{P} \tag{3}$$

where $\zeta$ is a binary variable that indicates if the generated text follows the output format in TAI; $\omega_0, ..., \omega_{m_p}$ are weighting parameters, which assign different levels of importance to the constraints.

In addition, the optimality-aware reward function is defined as:

$$\mathcal{R}_o^{\mathcal{P}}(\hat{x}_p) = \begin{cases} \alpha \frac{1}{1+M_o(\hat{x}_p)} & \text{if } \zeta \neq 0 \\ 0 & \text{if } \zeta = 0 \end{cases}, \forall p \in \mathcal{P} \qquad (4)$$

where $\alpha$ is a parameter that controls the relative importance between optimality and feasibility. We use the linear combination of $\mathcal{R}_f^{\mathcal{P}}$ and $\mathcal{R}_o^{\mathcal{P}}$ as the reward for reinforcement learning: $\mathcal{R}^{\mathcal{P}} = \mathcal{R}_f^{\mathcal{P}} + \mathcal{R}_o^{\mathcal{P}}$. The training process of FOARL is built upon group relative policy optimization (GRPO), which addresses inefficiencies of actor-critic methods [22]. Unlike the approaches that require a separate critic model, which often has a comparable size to the policy model, GRPO estimates the baseline directly from group-level rewards, significantly reducing the computational overhead while maintaining stable learning dynamics. More precisely, given a TAI $\phi(p)$, the algorithm samples $S$ solutions $\hat{X}_p = \{\hat{x}_{p,1}, \hat{x}_{p,2}, \ldots, \hat{x}_{p,S}\}$ using the old LLM policy model $\pi_{\theta,\text{old}}$, and update the current model $\pi_\theta$ (i.e., update the LoRA modules of the LLM) by maximizing the following objective function:

$$\mathcal{L}_{\text{FOARL}}(\theta) = \mathbb{E}_{\phi(p) \sim D^{\mathcal{P}}, \{\hat{x}_{p,i}\}_{i=1}^S \sim \pi_{\theta,\text{old}}(\hat{X}_p | \phi(p))}$$
$$\left[ \frac{1}{S} \sum_{i=1}^S \left( \min \left( r_i^{\text{ratio}} A_i, \text{clip} \left( r_i^{\text{ratio}}, 1-\epsilon, 1+\epsilon \right) A_i \right) - \beta D_{\text{KL}}(\pi_\theta \| \pi_{\text{ref}}) \right) \right], \quad (5)$$

where $D^{\mathcal{P}}$ is the training dataset for conducting reinforcement learning; $r_i^{\text{ratio}} = \frac{\pi_\theta(\hat{x}_{p,i} | \phi(p))}{\pi_{\theta,\text{old}}(\hat{x}_{p,i} | \phi(p))}$ measures how much the current policy differs from the old policy; $\epsilon$ and $\beta$ are hyperparameters; $D_{\text{KL}}(\pi_\theta \| \pi_{\text{ref}})$ is the Kullback–Leibler (KL) divergence between the current policy and the reference policy; $A_i = \frac{\mathcal{R}_i^{\mathcal{P}} - \text{mean}(\mathcal{R}_{\text{group}}^{\mathcal{P}})}{\text{std}(\mathcal{R}_{\text{group}}^{\mathcal{P}})}$ denotes the advantage of solution $\hat{x}_{p,i}$, computed using a group of rewards $\mathcal{R}_{\text{group}}^{\mathcal{P}} = \{\mathcal{R}_1^{\mathcal{P}}, \mathcal{R}_2^{\mathcal{P}}, \ldots, \mathcal{R}_S^{\mathcal{P}}\}$ corresponding to the sampled solutions within each group.

### 4.4 Best-of-N Inference

While SFT and FOARL provide LLMs with the capability to learn solution generation patterns, the autoregressive nature of these models inherently limits their ability to maintain globally optimal solutions throughout the generation process. Unlike traditional CO solvers that can explore multiple branches or backtrack upon reaching dead ends [50–52], autoregressive models commit to each decision and cannot easily revise earlier choices and evaluate constraint satisfaction, leading to suboptimal outcomes. To address this limitation, we employ Best-of-N (BoN) sampling during inference for test-time solution exploration. BoN sampling involves generating $N$ distinct solution candidates from the fine-tuned LLM and selecting the highest-quality feasible solution. Formally, given an instance $p$, we generate $N$ candidate solutions $\{\hat{x}_{p,1}, \hat{x}_{p,2}, \ldots, \hat{x}_{p,N}\}$ and select:

$$\hat{x}_p^* = \arg \min \{f_p(\hat{x}_{p,i}) \mid \hat{x}_{p,i} \in X_{C_p}, i \in \{1, 2, \ldots, N\}\} \qquad (6)$$

## 5 Experiments

We fine-tune Qwen2.5-7B [53] to serve as our dedicated end-to-end CO solvers. For SFT, we generate 500,000 instances per CO problem, and an additional set with at most 3,200 instances is used for FOARL. To improve training efficiency, we adopt the Unsloth framework [54] and apply LoRA for parameter-efficient fine-tuning. During inference, we set the temperature of the LLM to 0 for a single generation and use top-$p$ sampling ($p = 0.7$) for BoN inference ($N = 8$ solutions are generated by default). The optimality gaps are calculated using the domain-specific solvers of each problem, which are specified in Appendix A. The detailed experiment settings are provided in Appendix C.

### 5.1 Baselines

**General-purpose LLMs.** We first compare our method with general-purpose LLMs, including both closed-source and open-source models, all of which have significantly more parameters than our

Table 1: Evaluation of feasibility (fea.), optimality (opt.), and average time (Avg. Time) for different methods on the 7 studied CO problems. The best results are in bold.

| Method | TSP | | OP | | CVRP | | MIS | | MVC | | PFSP | | JSSP | | Avg. Time |
|---|---|---|---|---|---|---|---|---|---|---|---|---|---|---|---|
| | fea. | opt. | fea. | opt. | fea. | opt. | fea. | opt. | fea. | opt. | fea. | opt. | fea. | opt. | |
| **General-purpose Language Models** | | | | | | | | | | | | | | | |
| GPT-4o | 39% | $33.79\%_{\pm16.6}$ | 59% | $55.19\%_{\pm15.7}$ | 15% | $76.62\%_{\pm7.9}$ | 8% | $11.70\%_{\pm11.8}$ | 6% | $16.67\%_{\pm7.0}$ | 88% | $20.57\%_{\pm9.2}$ | 7% | $97.85\%_{\pm23.7}$ | 5.3s |
| GPT-4o-mini | 28% | $53.38\%_{\pm21.5}$ | 80% | $70.70\%_{\pm10.3}$ | 3% | $68.10\%_{\pm10.5}$ | 1% | $0.00\%_{\pm0}$ | 3% | $29.52\%_{\pm9.7}$ | 78% | $21.47\%_{\pm10.0}$ | 8% | $212\%_{\pm121}$ | 4.5s |
| Claude-Sonnet | 66% | $24.53\%_{\pm10.7}$ | 49% | $34.62\%_{\pm14.1}$ | 30% | $38.34\%_{\pm15.9}$ | 13% | $12.51\%_{\pm12.5}$ | 2% | $6.25\%_{\pm6.3}$ | 100% | $18.42\%_{\pm8.9}$ | 10% | $90.00\%_{\pm21.6}$ | 5.4s |
| Claude-Haiku | 45% | $38.60\%_{\pm16.9}$ | 26% | $51.26\%_{\pm14.9}$ | 4% | $61.48\%_{\pm12.1}$ | 2% | $23.33\%_{\pm6.7}$ | 6% | $43.01\%_{\pm23.3}$ | 73% | $20.58\%_{\pm7.6}$ | 9% | $95.51\%_{\pm22.2}$ | 5.1s |
| DeepSeek-V3 | 73% | $35.75\%_{\pm15.4}$ | 50% | $46.10\%_{\pm13.4}$ | 21% | $58.22\%_{\pm26.8}$ | 5% | $12.05\%_{\pm12.9}$ | 15% | $37.15\%_{\pm24.8}$ | 58% | $20.81\%_{\pm9.4}$ | 52% | $103.19\%_{\pm26.9}$ | 26.4s |
| Llama3.3-70B | 50% | $69.08\%_{\pm31.4}$ | 27% | $48.98\%_{\pm14.6}$ | 31% | $97.31\%_{\pm69.3}$ | 8% | $37.12\%_{\pm29.5}$ | 20% | $22.86\%_{\pm13.6}$ | 98% | $21.97\%_{\pm8.4}$ | 29% | $105.01\%_{\pm24.5}$ | 2.1s |
| Qwen2.5-72B | 20% | $36.89\%_{\pm34.6}$ | 32% | $49.36\%_{\pm16.6}$ | 61% | $180.91\%_{\pm105}$ | 14% | $29.56\%_{\pm16.2}$ | 5% | $63.20\%_{\pm30.4}$ | 98% | $21.13\%_{\pm8.2}$ | 53% | $103.90\%_{\pm61.7}$ | 12.5s |
| **Reasoning Models** | | | | | | | | | | | | | | | |
| GPT-o3-mini | 91% | $306\%_{\pm258}$ | 8% | $43.93\%_{\pm11.5}$ | 50% | $139\%_{\pm39.7}$ | 66% | $9.23\%_{\pm8.4}$ | 33% | $2.98\%_{\pm5.5}$ | 98% | $16.97\%_{\pm9.8}$ | 20% | $77.86\%_{\pm37.8}$ | 1.4m |
| GPT-o1 | 54% | $276\%_{\pm242}$ | 31% | $40.90\%_{\pm16.3}$ | 24% | $154\%_{\pm117}$ | 82% | $8.03\%_{\pm12.9}$ | 47% | $3.58\%_{\pm5.9}$ | 89% | $14.86\%_{\pm10.5}$ | 29% | $81.90\%_{\pm29.6}$ | 3.2m |
| DeepSeek-R1 | 48% | $70.99\%_{\pm23.1}$ | 60% | $40.54\%_{\pm13.7}$ | 26% | $30.46\%_{\pm18.1}$ | 41% | $1.60\%_{\pm3.6}$ | 38% | $4.17\%_{\pm6.3}$ | 100% | $16.65\%_{\pm8.1}$ | 5% | $26.29\%_{\pm8.5}$ | 6.5m |
| **Prompt Strategies** | | | | | | | | | | | | | | | |
| OPRO | 83% | $35.98\%_{\pm3.6}$ | 85% | $53.96\%_{\pm14.3}$ | 21% | $37.03\%_{\pm18.3}$ | 7% | $5.95\%_{\pm7.3}$ | 9% | $41.67\%_{\pm16.9}$ | 99% | $18.40\%_{\pm8.4}$ | 65% | $83.35\%_{\pm25.5}$ | 2.1m |
| LMEA | 77% | $265\%_{\pm131}$ | 48% | $66.18\%_{\pm10.5}$ | 24% | $61.24\%_{\pm19.0}$ | 5% | $25.0\%_{\pm14.2}$ | 13% | $34.22\%_{\pm16.3}$ | 98% | $14.31\%_{\pm7.1}$ | 44% | $83.19\%_{\pm23.9}$ | 5.3m |
| PHP | 84% | $33.84\%_{\pm14.6}$ | 43% | $36.08\%_{\pm15.1}$ | 33% | $58.11\%_{\pm26.4}$ | 5% | $11.67\%_{\pm9.1}$ | 13% | $19.84\%_{\pm10.01}$ | 92% | $17.23\%_{\pm8.1}$ | 56% | $104.04\%_{\pm29.2}$ | 1.6m |
| SGE | 98% | $29.66\%_{\pm43.3}$ | 93% | $24.49\%_{\pm38.4}$ | 84% | $36.14\%_{\pm59.2}$ | 92% | $3.62\%_{\pm7.6}$ | 94% | $3.83\%_{\pm7.4}$ | 95% | $4.48\%_{\pm7.4}$ | 87% | $38.58\%_{\pm49.2}$ | 3.6m |
| **Ours** | | | | | | | | | | | | | | | |
| SFT | 89% | $2.30\%_{\pm1.9}$ | 54% | $2.32\%_{\pm2.6}$ | 59% | $6.02\%_{\pm3.9}$ | 80% | $1.71\%_{\pm3.9}$ | 98% | $2.41\%_{\pm3.3}$ | 100% | $2.22\%_{\pm1.9}$ | 100% | $11.01\%_{\pm7.9}$ | 5.6s |
| SFT+RL | 91% | $2.32\%_{\pm2.2}$ | 92% | $4.25\%_{\pm2.9}$ | 80% | $8.27\%_{\pm5.6}$ | 83% | $1.34\%_{\pm3.3}$ | 98% | $2.39\%_{\pm3.2}$ | 100% | $2.12\%_{\pm1.8}$ | 100% | $10.94\%_{\pm7.3}$ | 5.6s |
| SFT+RL+BoN | **100%** | $\mathbf{1.07\%_{\pm0.9}}$ | **100%** | $\mathbf{1.85\%_{\pm1.7}}$ | **100%** | $\mathbf{4.53\%_{\pm3.5}}$ | **94%** | $\mathbf{1.04\%_{\pm3.4}}$ | **100%** | $\mathbf{1.29\%_{\pm2.2}}$ | **100%** | $\mathbf{1.03\%_{\pm1.1}}$ | **100%** | $\mathbf{8.20\%_{\pm6.3}}$ | 9.8s |

fine-tuned model. The used models include OpenAI GPT-4o, GPT-4o-mini, Anthropic Claude-3.7-Sonnet-20250219, Claude-3.5-Haiku, DeepSeek-V3-671B, LLaMA-3.3-70B, and Qwen2.5-72B.

**Reasoning LLMs.** With the rapid development of reasoning models, many previously challenging mathematical tasks have become tractable. Therefore, we also include state-of-the-art reasoning models for comparison, such as OpenAI GPT-o3-mini, GPT-o1, and DeepSeek-R1.

**Prompt Strategies.** We consider representative LLM-based optimization methods for comparison. These include Optimization by PROmpting (OPRO) [19], which steers solution generation using verbal gradients; LMEA (LLM-driven Evolutionary Algorithm) [34], which treats LLMs as evolutionary optimizers by selecting parent solutions from the current population and applying crossover and mutation to produce offspring; and progressive-hint prompting (PHP) [55], which scales test-time performance by incorporating previous outputs as hints to guide subsequent reasoning. We also evaluate the methods that leverage LLMs for program generation, such as self-guiding exploration (SGE) [56]. All prompt-based baselines are implemented using DeepSeek-V3-671B for consistency.

**Domain-specific methods.** We employ OR-Tools [57], a heuristic solver that can solve TSP and CVRP. We also use ant colony optimization (ACO) [58], serving as the metaheuristic to solve the routing problems. Some heuristics are also introduced, such as the nearest neighbor (NN) and farthest insertion (FI) for TSP, the greedy method, greedy insertion, and Tsili algorithm [59] for OP, the sweeping and parallel savings (PS) algorithm for CVRP, the greedy minimum degree (Greedy) and degree-based add (Degree) heuristic for MIS, the Approx method [60], greedy maximum degree (Greedy) and degree-based removal (Degree) heuristic for MVC, the Plamer's [61] and Nawaz, Enscore, and Ham (NEH) heuristic [62] for PFSP, and commonly used dispatching rules for JSSP, such as shortest processing time (SPT), first in first out (FIFO), and apparent tardiness cost (ATC).

More details of the baselines are elaborated in Appendix D. Our code and data are publicly available at `https://github.com/Summer142857/LLMCoSolver`.

## 5.2 Main Results

First, we compare our method against a wide range of advanced LLMs and LLM-based optimization baselines, with the results summarized in Table 1, while the solving times of each problem are provided in Appendix E.1. It is observed that general-purpose LLMs generally fail to produce feasible solutions for most CO tasks. An exception is the PFSP task, where models only need to generate a permutation of a few job indices, which is relatively simpler compared to other problems.

Although some general-purpose LLMs and advanced reasoning models can occasionally generate feasible solutions for certain tasks (e.g., GPT-o3-mini for TSP and Claude-3.7-Sonnet for PFSP), they exhibit substantial gaps in solution optimality. Moreover, the inference time of reasoning models is

Table 2: Performance comparison on CO problems with different scales of graphs.

| Method | Small graphs | | | | Medium graphs | | | | Large graphs | | | |
|---|---|---|---|---|---|---|---|---|---|---|---|---|
| | Gap | Gap@1 | Gap@5 | Gap@10 | Gap | Gap@1 | Gap@5 | Gap@10 | Gap | Gap@1 | Gap@5 | Gap@10 |
| **TSP** | | | | | | | | | | | | |
| OR-Tools | 0.82% | 76% | 96% | 99% | 2.59% | 28% | 86% | 99% | 3.59% | 12% | 80% | 99% |
| NN | 19.36% | 3% | 5% | 19% | 24.13% | 0% | 0% | 1% | 26.19% | 0% | 0% | 2% |
| FI | 2.27% | 51% | 82% | 96% | 4.41% | 8% | 64% | 95% | 4.86% | 2% | 53% | 99% |
| ACO | 1.98% | 48% | 88% | 100% | 17.98% | 0% | 1% | 6% | 36.69% | 0% | 0% | 0% |
| Ours | 0.14% | 96% | 100% | 100% | 0.70% | 74% | 100% | 100% | 1.34% | 44% | 100% | 100% |
| **OP** | | | | | | | | | | | | |
| Greedy | 16.06% | 4% | 8% | 25% | 18.91% | 0% | 0% | 5% | 20.30% | 0% | 0% | 3% |
| GI | 9.07% | 20% | 37% | 63% | 11.91% | 0% | 12% | 51% | 13.63% | 0% | 7% | 32% |
| Tsili | 3.85% | 21% | 68% | 96% | 9.54% | 0% | 2% | 55% | 13.80% | 0% | 0% | 8% |
| ACO | 3.49% | 30% | 76% | 94% | 6.24% | 1% | 35% | 89% | 7.95% | 0% | 15% | 74% |
| Ours | 1.47% | 54% | 95% | 99% | 2.04% | 26% | 96% | 100% | 2.10% | 27% | 96% | 99% |
| **CVRP** | | | | | | | | | | | | |
| OR-Tools | 3.60% | 45% | 69% | 93% | 7.87% | 3% | 24% | 72% | 8.84% | 0% | 15% | 71% |
| Sweep | 18.36% | 8% | 18% | 32% | 20.59% | 0% | 1% | 8% | 22.07% | 0% | 0% | 3% |
| PS | 3.95% | 24% | 72% | 93% | 5.67% | 2% | 50% | 89% | 6.12% | 0% | 41% | 89% |
| ACO | 2.52% | 44% | 81% | 96% | 17.07% | 0% | 2% | 14% | 29.49% | 0% | 0% | 0% |
| Ours | 1.70% | 52% | 90% | 97% | 4.57% | 8% | 59% | 98% | 7.24% | 1% | 19% | 84% |
| **MIS** | | | | | | | | | | | | |
| Degree | 5.89% | 57% | 57% | 67% | 7.52% | 32% | 42% | 64% | 9.61% | 20% | 33% | 59% |
| Greedy | 2.56% | 84% | 84% | 86% | 2.79% | 67% | 71% | 90% | 3.36% | 53% | 60% | 83% |
| Ours | 0.38% | 97% | 98% | 98% | 1.05% | 86% | 87% | 94% | 2.29% | 47% | 57% | 65% |
| **MVC** | | | | | | | | | | | | |
| Approx | 49.79% | 0% | 0% | 0% | 39.21% | 0% | 0% | 0% | 35.35% | 0% | 0% | 0% |
| Greedy | 2.80% | 71% | 72% | 89% | 2.62% | 44% | 79% | 99% | 2.21% | 33% | 89% | 100% |
| Degree | 3.53% | 63% | 63% | 86% | 2.78% | 44% | 78% | 98% | 2.63% | 26% | 83% | 99% |
| Ours | 0.48% | 93% | 93% | 100% | 1.25% | 66% | 94% | 100% | 2.35% | 38% | 87% | 100% |
| **PFSP** | | | | | | | | | | | | |
| Palmer's | 30.52% | 0% | 0% | 1% | 30.41% | 0% | 0% | 0% | 30.68% | 0% | 0% | 0% |
| NEH | 1.33% | 53% | 97% | 99% | 2.78% | 11% | 90% | 100% | 3.56% | 0% | 88% | 100% |
| Ours | 0.25% | 85% | 100% | 100% | 1.16% | 40% | 100% | 100% | 2.62% | 6% | 99% | 100% |
| **JSSP** | | | | | | | | | | | | |
| SPT | 19.58% | 2% | 4% | 17% | 25.32% | 0% | 0% | 1% | 27.35% | 0% | 0% | 0% |
| FIFO | 24.38% | 2% | 4% | 12% | 32.97% | 0% | 0% | 1% | 39.00% | 0% | 0% | 0% |
| ATC | 20.71% | 0% | 12% | 15% | 24.30% | 0% | 0% | 1% | 27.99% | 0% | 0% | 0% |
| Ours | 2.86% | 32% | 79% | 98% | 9.56% | 0% | 8% | 60% | 16.25% | 0% | 0% | 4% |

significantly longer, as they attempt to solve CO problems through extended step-by-step thinking. While such approaches may work for very small-scale instances (e.g., fewer than 10 nodes), it becomes impractical to reason clearly and effectively as instance size grows. For example, we present the CoT of DeepSeek-R1 for a TSP instance in Appendix E.2, where we find that the CoT is lengthy and ineffective. This causes the LLM to get stuck multiple times and cannot find a good solution.

Although iterative prompting strategies (e.g., OPRO and PHP) can moderately improve performance, they still struggle to produce competitive results and remain far from practical deployment. Leveraging LLMs to generate code for (meta)heuristic algorithms, as done in SGE, enables feasible solutions for most instances. However, their solution quality remains limited, mainly because the generated algorithms are highly sensitive to hyperparameter tuning—a process that requires significant human expertise and cannot be reliably automated by LLMs alone. Meanwhile, these prompting strategies rely on iterative reasoning or code execution, leading to long solving time compared to our method.

In contrast, our proposed SFT approach enables the LLM to generate mostly feasible solutions with a relatively small optimality gap. By further combining SFT with FOARL and BoN inference, our method achieves a 100% feasibility rate across the test sets of TSP, OP, CVRP, MVC, PFSP, and JSSP. Both feasibility and optimality performance significantly surpass all other baselines, including those based on much larger models than our fine-tuned 7B model. Additionally, as shown in Appendix E.3, our models also outperform a recent language-based CO solver proposed in [14], which integrates the LLMs with an NCO model, across most of the evaluated tasks, demonstrating its effectiveness.

We also compare against the LLM-based model-then-solve approaches. As demonstrated in Appendix E.4, our method can outperform these approaches, particularly when dealing with large instances.

### 5.3 Comparison with Domain-specific Methods

A variety of domain-specific algorithms have been developed for individual CO problems, typically based on (meta)heuristic strategies, and often characterized by distinct algorithmic structures. We compare our LLM-based approach (i.e., the model trained by SFT+RL and with a BoN sampling) to these methods, with results summarized in Table 2. The evaluation is conducted across multiple problem scales, with 100 instances used for each. Specifically, small, medium, and large graph instances are defined as follows: for scheduling problems, 5×5–10×10, 10×10–15×15, and 15×15–20×20;

for other problems, 10–30, 40–60, and 70–100 nodes, respectively. We also introduce Gap@K, the percentage of instances solved with an optimality gap below K%. As shown in the table, our model can outperform the simple domain-specific algorithms, especially on smaller graphs, indicating its strong potential for solving CO problems described by language.

Meanwhile, we also provide the comparative results between our method and the state-of-the-art heuristic solvers and classical exact solvers (i.e., Gurobi), which are provided in Appendix E.5. It is shown that our method can achieve better performance on routing tasks than Gurobi under the same solving time budget. We also discuss the advantages of our method (i.e., ease of use, language-driven interaction, and generalizability) compared to these classical solvers in Appendix E.5.

Moreover, as it is possible to further enhance the performance of the LLM solver through scaling test-time exploration, we increase $N$ for BoN sampling and report the results in Appendix E.6, where we find that the instances with large graphs can be better solved with a higher $N$. We also examine the impact of heuristic features in the TAI through an ablation study, as presented in Appendix E.7.

## 5.4 Role of Reinforcement Learning

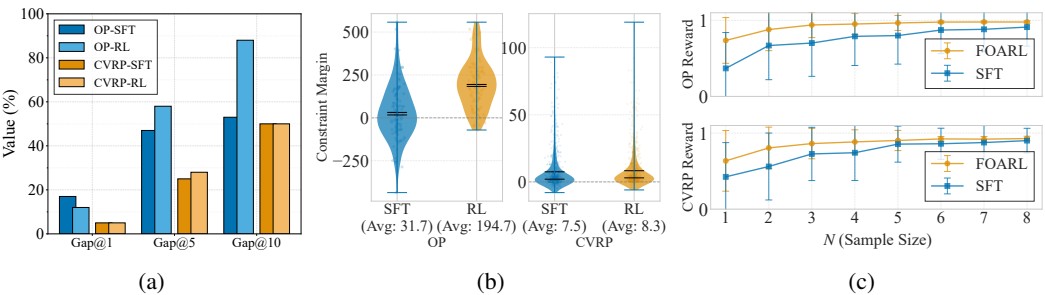

(a)                                    (b)                                    (c)

Figure 3: The role of FOARL. (a) RL does not necessarily cause a performance drop in Gap@K; (b) Distribution of constraint margins before and after applying FOARL. Positive values indicate feasible solutions, while negative values reflect infeasibility due to violations of inequality constraints; (c) The reward comparison between policies with different $N$ for BoN sampling.

According to Table 1, FOARL significantly improves feasibility in tasks such as OP and CVRP, while incurring minor losses in optimality. This is primarily because FOARL enables the LLM to generate feasible solutions for more instances, rather than degrading the quality of solutions on previously solved ones. As illustrated in Figure 3 (a), The Gap@K metric does not exhibit a noteworthy decrease after applying RL. Furthermore, we observe slight improvements in optimality for tasks where the SFT model already exhibits high feasibility. To better understand the contribution of RL, we analyze its dual role using two representative tasks (i.e., OP and CVRP) as case studies, as detailed below.

**Constraint relaxation operator.** FOARL helps enforce feasibility by mitigating violations of inequality constraints, which are common in problems like OP and CVRP. We define the constraint margin as: 1) the difference between actual travel distance and the distance limit in OP, 2) the difference between served demand and vehicle capacity in CVRP, and visualize them for solutions generated by both SFT and RL in Figure 3 (b). The increase in constraint margins of RL-trained policies indicates improved feasibility and correction of the over-greedy behavior introduced by SFT.

**Sampling efficiency improver.** More generally, the RL process improves sampling efficiency during inference. While it is possible to enhance the SFT policy through increased test-time exploration, such as scaling BoN sampling, RL allows for achieving better performance with fewer samples, thereby improving efficiency and reducing test-time computational cost. Specifically, we vary the scale of BoN sampling for both SFT policy and RL policy, evaluating them on instances with the large graphs. The change of reward $\mathcal{R}_o^{\mathcal{P}}$ (defined by Equation 4), which can indicate both feasibility (e.g., $\mathcal{R}_o^{\mathcal{P}} = 0$ if a solution is infeasible) and optimality, is illustrated in Figure 3 (c). We observe that models trained with RL require smaller $N$ to reach comparable performance levels as their SFT-only counterparts with larger $N$. This also aligns with the conclusion of a recent work [63]. Since computational efficiency is critical in some real-time or resource-constrained decision-making scenarios, incorporating RL to improve inference efficiency is a valuable and practical enhancement.

Table 3: Comparison study across TA instances. *: Results are drawn from the original literature.

| | TA 15x15 | | TA 20x15 | | TA 20x20 | |
|---|---|---|---|---|---|---|
| | obj.↓ | opt.↓ | obj.↓ | opt.↓ | obj.↓ | opt.↓ |
| Best-known solution | 1228.9 | - | 1364.9 | - | 1617.3 | - |
| SPT | 1546.1 | 25.81% | 1813.5 | 32.87% | 2067.2 | 27.81% |
| FIFO | 1657.4 | 34.87% | 2008.4 | 47.15% | 2297.1 | 42.03% |
| ATC | 1586.8 | 29.12% | 1794.3 | 31.46% | 2114.1 | 30.72% |
| GA* 100x100 | 1583.8 | 28.88% | 1847.1 | 35.23% | 2304.2 | 42.47% |
| L2D greedy | 1522.3 | 23.88% | 1813.4 | 32.86% | 2130.2 | 31.21% |
| L2D sample | 1543.8 | 25.62% | 1779.7 | 30.39% | 2134.2 | 31.96% |
| StarJob* | - | 19.68% | - | 26.91% | - | 33.12% |
| Ours ($N = 8$) | 1403.6 | 14.22% | 1653.7 | 21.16% | 1994.2 | 23.30% |
| Ours ($N = 64$) | 1368.2 | 11.34% | 1600.5 | 17.26% | 1940.9 | 20.01% |

## 5.5 Unified CO solver

Recent research progress in neural network-based solvers is being made towards the establishment of a unified model, which can solve various CO problems. The existing unified models usually use task-specific encoders and decoders [44] or problem reduction [64], limiting their extensibility. By comparison, our method provides a more general approach using natural language as the interface. We fine-tune the LLM by learning to solve the seven CO problems together, and present the results in Appendix E.8, where we observe that a single LLM solver can also solve various CO problems.

## 5.6 Versatility and Generalizability

To show the versatility of our method, we additionally evaluate it using different LLMs (i.e., Llama-3.1-8B and Gemma-2-9B), as detailed in Appendix E.9. We evaluate the out-of-distribution (OOD) performance of the trained LLMs for routing problems, as presented in Appendix E.10. There, we demonstrate that the LLM solvers can also generalize to instances with OOD distributions.

## 5.7 Benchmarking Performance

To further demonstrate the generalizability, we evaluate the fine-tuned JSSP solver on Taillard (TA) benchmark [65]. We test on the instances with 15 jobs and 15 machines ($15 \times 15$), 20 jobs and 15 machines ($20 \times 15$), and 20 jobs and 20 machines ($20 \times 20$). In addition to the dispatching rules, such as SPT, FIFO, and ATC, we also compare with the genetic algorithm (GA) (with a population size of 100 and 100 generations) [66], the L2D method [67], and StarJob [48], which solves JSSP using an LLM fine-tuned through SFT. The results are compared in Table 3. When all solutions are feasible, our model (SFT+RL) outperforms all heuristic and learning-based baselines, with performance further improved by increasing $N$ to 64 to enhance test-time exploration. Meanwhile, we also compare our method with other LLM-based methods on TSPLib, and the result is presented in Appendix E.11.

## 6 Conclusion

The paper investigates how LLMs can be fine-tuned to serve as end-to-end CO solvers, filling the gap in the literature on this topic. Taking CO problem solving as a language generation task, we propose a two-stage fine-tuning framework, incorporating both SFT and FOARL, to enable the LLMs to solve CO problems in an end-to-end manner. The tuned LLM solver achieves the average optimality gaps of 1.03–8.20% for different problems. Following the traditional heuristic solvers and NCO solvers, our method, mediated by language, promises to be a new paradigm for CO, which provides a more general end-to-end approach and reduces human expertise reliance. One major limitation of our approach, shared by most LLM-based methods, is that its solving efficiency is not yet on par with traditional lightweight heuristics. Future directions include: 1) introducing efficient fine-tuning and inference strategies to solve larger-scale CO problems; 2) exploring more efficient and effective textual input representation, i.e., TAIs, to enhance learning performance; 3) integrating the LLM solver with heuristics to further address infeasibility and improve optimality.

## Acknowledgments and Disclosure of Funding

This research is supported by the National Research Foundation, Singapore under its AI Singapore Programme (AISG Award No: AISG3-RP-2022-031). This work has also made use of resources and expertise provided by SURF Experimental Technologies Platform, which is part of the SURF cooperative in the Netherlands (No.EINF-13168).

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

# Large Language Models as End-to-end Combinatorial Optimization Solvers
## (Appendices)

## A  Specification of the Studied Problems

This section provides detailed definitions of the seven combinatorial optimization problems studied in this paper, which also includes their instance generation procedures, textual instance representations (i.e., TAIs), and the FOARL reward settings tailored to each task.

Note that we incorporate problem-specific heuristic features in the TAIs in order to facilitate effective learning while preserving the generality and efficiency of our language-based approach. These heuristic features are based on classical greedy principles, and are tailored to each CO problem but follow a consistent philosophy: they should be 1) computationally inexpensive, 2) representative of well-known heuristic behavior, and 3) easily expressible in natural language. For example, in routing problems like TSP and CVRP, we include the top-$k$ nearest neighbors for each node along with their distances, inspired by nearest-neighbor heuristics. In scheduling problems like PFSP, we select the top-$k$ jobs with the shortest processing times, aligning with the common dispatching rules such as Shortest Processing Time. Importantly, because they can be concisely described in text, they are naturally suited to the language input format required by LLMs. This contrasts with more complex or learned heuristics, which often require code-level specifications or internal neural representations that are difficult to translate into promptable natural language. By embedding these features into the input, we provide informative, domain-aligned cues that help the LLM better solve different CO problems.

### A.1  Traveling Salesman Problem (TSP)

#### A.1.1  Problem Definition

The TSP involves finding the shortest possible route that visits each city exactly once and returns to the origin. Formally, given a set of nodes $V = \{0, 1, 2, \ldots, n-1\}$ and a distance matrix $d$ where $d_{ij}$ represents the distance between nodes $i$ and $j$, we introduce binary decision variables $x_{ij}$ where $x_{ij} = 1$ if the tour includes a direct trip from node $i$ to node $j$, and $x_{ij} = 0$ otherwise. The mathematical model of TSP is as:

$$\text{minimize} \quad \sum_{i=0}^{n-1} \sum_{j=0, j \neq i}^{n-1} d_{ij} x_{ij} \tag{7}$$

$$\text{subject to} \quad \sum_{j=0, j \neq i}^{n-1} x_{ij} = 1 \quad \forall i \in V \tag{8}$$

$$\sum_{i=0, i \neq j}^{n-1} x_{ij} = 1 \quad \forall j \in V \tag{9}$$

$$\sum_{i,j \in S, i \neq j} x_{ij} \leq |S| - 1 \quad \forall S \subset V, 2 \leq |S| \leq n-1 \tag{10}$$

$$x_{ij} \in \{0, 1\} \quad \forall i, j \in V, i \neq j \tag{11}$$

where $S$ represents any proper subset of nodes. Constraints 1 and 2 ensure that each node is visited exactly once, while constraint 3 (subtour elimination constraints) ensures the solution forms a single tour covering all nodes rather than multiple disconnected subtours.

#### A.1.2  Instance Generation

We generate cross-size and cross-distribution instances for the TSP by randomly sampling node numbers from $[10, 100] \cap \mathbb{Z}$ and node coordinate distributions from $[1, 1000] \cap \mathbb{Z}$, following a uniform distribution and two Gaussian mixture (GM) distributions. The GM distribution is parameterized

by cluster $c$ and scale $l$, and we use $c = 2, l = 5$ and $c = 3, l = 10$ respectively. The overall generation process follows the existing work [68]. We also calculate the top-2 nearest neighbors while transforming the numerical instances to TAIs using KD-Tree. The solutions (labels) of the generated instances are obtained by the LKH-3 solver, which can produce optimal solutions for TSP with node number from 10 to 100. An example TAI used for SFT is shown below:

---

**Example TAI of TSP**

### Instruction: Solve the Traveling Salesman Problem (TSP) for the given list of 81 cities. Each city is represented as a node with coordinates (x, y). Identify the shortest route that visits every city exactly once and returns to the starting city. The input includes city coordinates, the 2 nearest neighbors for each city, and their respective distances. Provide the solution in the following format: 1. Route: List the nodes in the order they are visited. 2. Objective: The objective value (total travel distance).
### Input: Node 0, coordinates: [374, 950], neighbors: [69: 23.7, 12: 184.3]; Node 1, coordinates: [731, 598], neighbors: [43: 39.1, 46: 47.0]; Node 2, coordinates: [156, 155], neighbors: [7: 37.5, 28: 78.9]; Node 3, coordinates: [58, 866], neighbors: [21: 49.2, 36: 73.6]; Node 4, coordinates: [601, 708], neighbors: [68: 83.2, 78: 88.6]; Node 5, coordinates: [20, 969], neighbors: [16: 49.7, 34: 56.6]...
### Output: Route: [0, 27, 57, 60, 63, 26, 56, 17, 25, 40, 59, 44, 31, 67, 77, 70, 52, 75, 6, 58, 35, 38, 14, 10, 15, 24, 65, 39, 61, 18, 41, 72, 54, 55, 7, 2, 49, 28, 74, 29, 66, 62, 11, 42, 30, 9, 71, 48, 73, 19, 47, 46, 1, 43, 37, 78, 4, 68, 12, 53, 79, 22, 80, 51, 23, 8, 32, 13, 76, 20, 64, 50, 45, 33, 36, 3, 21, 5, 16, 34, 69, 0], Objective: 6833.347

---

### A.1.3 Reward Function

We use rule-based rewards to represent constraint satisfaction for all the CO problems. According to the definition of the TSP, we define the feasibility reward as $\mathcal{R}_f = \omega_0 \zeta + \omega_1 c_1 + \omega_2 c_2$, where $\omega_0 = 0.2$, $\omega_1 = 0.5$, and $\omega_2 = 0.3$. Here, $c_1 = 1$ if all nodes are visited exactly once (otherwise 0), and $c_2 = 1$ if the route returns to the starting point (otherwise 0). $\zeta$ represents whether the route can be extracted using the specific format (which is shown in the example TAI) from the generated textual output. For simplicity, we omit the notation of the problem class $P$ and the instance $p$.

### A.2 Orienteering Problem (OP)

### A.2.1 Problem Definition

The OP involves finding a path from a starting point, visiting a subset of nodes to maximize the total collected prize while keeping the total distance within a given limit. Formally, given a set of nodes $V = \{0, 1, 2, \ldots, n-1\}$ with prizes $s_i$ for each node $i$, a distance matrix $d$ where $d_{ij}$ represents the distance between nodes $i$ and $j$, and a distance limit $B$, we introduce binary decision variables $x_i$ where $x_i = 1$ if node $i$ is visited, and $x_i = 0$ otherwise, and binary decision variables $y_{ij}$ where $y_{ij} = 1$ if the path includes a direct trip from node $i$ to node $j$. The mathematical model of OP is as:

$$\text{maximize} \quad \sum_{i=1}^{n-2} s_i x_i \tag{12}$$

$$\text{subject to} \quad \sum_{j=1}^{n-1} y_{0j} = 1 \tag{13}$$

$$\sum_{j=0, j \neq i}^{n-1} y_{ij} = x_i \quad \forall i \in V \setminus \{n-1\} \tag{14}$$

$$\sum_{i=0, i \neq j}^{n-1} y_{ij} = x_j \quad \forall j \in V \setminus \{0\} \tag{15}$$

$$\sum_{i=0}^{n-1} \sum_{j=0, j \neq i}^{n-1} d_{ij} y_{ij} \leq B \tag{16}$$

$$\sum_{i,j \in S, i \neq j} y_{ij} \leq \sum_{i \in S} x_i - x_k \quad \forall S \subset V, k \in S, 2 \leq |S| \leq n-1 \tag{17}$$

Constraint 1 ensures that the path starts at node 0. Constraints 2 and 3 ensure flow conservation at each visited node. Constraint 4 ensures the total distance does not exceed the distance limit $B$. Constraint 5 (subtour elimination constraint) ensures the solution forms a single path rather than disconnected subtours. The final two constraints define the binary decision variables.

### A.2.2 Instance Generation

The size and node distributions of the generated instances are the same as TSP. The prize of each node is randomly sampled from $[1, 10] \cap \mathbb{Z}$. As for the route limit, we firstly use a simple heuristic, i.e., the nearest neighbor method, to obtain the approximated route length $L_T$ for the TSP that corresponds to the OP instance, and randomly sample a float number from $[0.5L_T, 0.7L_T]$ as the value of $B$. For simplicity, we specify that the route does not have to return to the depot. The optimal solutions are obtained by a recent state-of-the-art solver for the OP based on a genetic algorithm [69]. Below is an example TAI for the OP task:

> **Example TAI of OP**
>
> ### Instruction: Solve the Orienteering Problem with 23 nodes. Each node has (x, y) coordinates and a prize for visiting it. You must plan a route that starts at depot 0, collecting the maximum total prize possible, subject to a maximum route length T = 2682.5. You may visit a subset of nodes, but the total distance traveled must not exceed T. The input includes city coordinates, the 2 nearest neighbors for each city, and their respective distances. Provide the solution in the following format: 1. Route: The ordered list of visited nodes. 2. Objective: The objective value (summation of the collecting prizes).
> ### Input: Node 0, coordinates: [0, 871], prize: 0, neighbors: [2: 84.6, 5: 142.2]; Node 1, coordinates: [208, 957], prize: 5, neighbors: [4: 81.0, 3: 82.2]; Node 2, coordinates: [46, 800], prize: 5, neighbors: [0: 84.6, 5: 150.9]; Node 3, coordinates: [138, 1000], prize: 2, neighbors: [4: 44.4, 5: 76.2]; Node 4, coordinates: [127, 957], prize: 10, neighbors: [5: 33.4, 3: 44.4]; Node 5, coordinates: [132, 924], prize: 6, neighbors: [4: 33.4, 3: 76.2]...
> ### Output: Route: [0, 2, 6, 8, 11, 19, 22, 18, 16, 14, 10, 13, 12, 9, 1, 5], Objective: 98.00

### A.2.3 Reward Function

The feasibility reward of the orienteering problem is defined as $\mathcal{R}_f = \omega_0 \zeta + \omega_1 c_1 + \omega_2 c_2 + \omega_3 c_3$, where $\omega_0 = 0.2$, $\omega_1 = 0.1$, $\omega_1 = 0.2$, and $\omega_3 = 0.5$. Specifically, $c_1 = 1$ if the route starts from the depot (otherwise 0); $c_2 = 1$ if each node is visited at most once (otherwise 0); $c_3 = 1$ if the total travel distance is within the distance limit $B$ (otherwise 0). The LLM can easily violate the distance

limit constraint, so we assign a higher weight to it during reinforcement learning. The optimality reward of OP is $\mathcal{R}_o^{\mathcal{P}}(\hat{x}_p) = \frac{\hat{x}_p}{x_p^*}$ because we try to maximize the collected prizes.

### A.3 Capacitated Vehicle Routing Problem (CVRP)

#### A.3.1 Problem Definition

The CVRP involves finding optimal routes for a fleet of vehicles to deliver goods from a depot to customers, where each vehicle has a limited capacity (we use homogeneous capacity for all the vehicles for simplicity). Formally, given a set of nodes $V = \{0, 1, 2, \ldots, n\}$ where node 0 represents the depot, a distance matrix $d$ where $d_{ij}$ represents the distance between nodes $i$ and $j$, customer demands $q_i$ for each node $i \in V \setminus \{0\}$, a fleet of $K$ identical vehicles each with capacity $Q$, we introduce binary decision variables $x_{ijk}$ where $x_{ijk} = 1$ if vehicle $k$ travels directly from node $i$ to node $j$, and $x_{ijk} = 0$ otherwise. The mathematical model of CVRP is as:

$$\text{minimize} \quad \sum_{k=1}^{K} \sum_{i=0}^{n} \sum_{j=0, j \neq i}^{n} d_{ij} x_{ijk} \tag{18}$$

$$\text{subject to} \quad \sum_{k=1}^{K} \sum_{j=1}^{n} x_{ijk} = 1 \quad \forall i \in V \setminus \{0\} \tag{19}$$

$$\sum_{j=1}^{n} x_{0jk} = 1 \quad \forall k \in \{1, \ldots, K\} \tag{20}$$

$$\sum_{i=0}^{n} x_{ihk} = \sum_{j=0}^{n} x_{hjk} \quad \forall h \in V, \forall k \in \{1, \ldots, K\} \tag{21}$$

$$\sum_{i=0}^{n} x_{i0k} = 1 \quad \forall k \in \{1, \ldots, K\} \tag{22}$$

$$\sum_{i \in V \setminus \{0\}} \sum_{j \in V} q_i x_{ijk} \leq Q \quad \forall k \in \{1, \ldots, K\} \tag{23}$$

$$\sum_{i,j \in S, i \neq j} x_{ijk} \leq |S| - 1 \quad \forall S \subset V \setminus \{0\}, |S| \geq 2, \forall k \in \{1, \ldots, K\} \tag{24}$$

$$x_{ijk} \in \{0, 1\} \quad \forall i, j \in V, i \neq j, \forall k \in \{1, \ldots, K\} \tag{25}$$

Constraint 1 ensures that each customer is visited exactly once by exactly one vehicle. Constraints 2 and 4 ensure that each vehicle starts and ends at the depot. Constraint 3 ensures flow conservation at each node (a vehicle that enters a node must also leave it). Constraint 5 ensures that the total demand served by each vehicle does not exceed its capacity $Q$. Constraint 6 (subtour elimination constraints) ensures that each vehicle's route forms a single tour connected to the depot rather than disconnected subtours. The final constraint defines the binary decision variables.

#### A.3.2 Instance Generation

The size and node distributions of the generated instances are also the same as TSP (10-100 nodes with 3 different distributions). The values of customer demands $q_i$ are randomly sampled from $[1, 10] \cap \mathbb{Z}$, and the capacity of each vehicle is set to the average value among all the demands. The near-optimal solutions used for fine-tuning are also produced by the LKH-3 solver. Below is an example TAI for the CVRP:

> **Example TAI of CVRP**
>
> ### Instruction: Solve the Capacitated Vehicle Routing Problem (CVRP) with 45 customers and 1 depot (node 0). Each customer node has a demand. All vehicles have the same capacity of 111. You must assign each customer to exactly one route and ensure that the sum of demands on each route does not exceed the vehicle capacity. Minimize the total distance traveled. The input includes city coordinates, the 2 nearest neighbors for each city, and their respective distances. Provide the solution in the following format: 1. Route: A list of routes, each route as an ordered list of visited nodes (start/end at the depot). 2. Objective: The total distance of all routes.
> ### Input: Node 0, coordinates: [428, 688], demand: 0, neighbors: [6: 42.6, 14: 53.1]; Node 1, coordinates: [58, 915], demand: 9, neighbors: [15: 35.4, 17: 46.6]; Node 2, coordinates: [442, 239], demand: 3, neighbors: [44: 159.4, 27: 170.8]; Node 3, coordinates: [93, 182], demand: 9, neighbors: [8: 89.8, 10: 129.0]; Node 4, coordinates: [934, 638], demand: 7, neighbors: [21: 67.0, 40: 102.1]; Node 5, coordinates: [516, 657], demand: 4, neighbors: [14: 52.6, 0: 93.3]...
> ### Output: Routes: [[0, 6, 35, 1, 15, 17, 12, 7, 10, 3, 8, 27, 45, 9, 28, 23, 2, 32, 39, 41, 0], [0, 5, 37, 19, 22, 21, 40, 4, 20, 24, 25, 0], [0, 43, 44, 29, 26, 42, 38, 34, 11, 16, 13, 33, 31, 30, 36, 18, 14, 0]], Objective: 6643.76

### A.3.3 Reward Function

The feasibility reward of the CVRP is similar to OP, which is defined as $\mathcal{R}_f = \omega_0\zeta + \omega_1 c_1 + \omega_2 c_2 + \omega_3 c_3$, where $\omega_0 = 0.2$, $\omega_1 = 0.1$, $\omega_1 = 0.1$, and $\omega_3 = 0.6$. Specifically, $c_1 = 1$ if the route starts from the depot (otherwise 0); $c_2 = 1$ if all customer nodes are visited exactly once; $c_3 = 1$ if the capacity constraint is satisfied (the total demand served by each vehicle does not exceed $Q$) (otherwise 0).

## A.4 Maximal Independent Set (MIS)

### A.4.1 Problem Definition

The MIS problem involves finding the largest subset of vertices in a graph such that no two vertices in the subset are adjacent. Formally, given an undirected graph $G = (V, E)$ where $V = \{1, 2, \ldots, n\}$ is the set of vertices and $E$ is the set of edges, we introduce binary decision variables $x_i$ where $x_i = 1$ if vertex $i$ is included in the independent set, and $x_i = 0$ otherwise. The mathematical model is as:

$$\text{maximize} \quad \sum_{i=1}^{n} x_i \tag{26}$$

$$\text{subject to} \quad x_i + x_j \leq 1 \quad \forall (i, j) \in E \tag{27}$$

$$x_i \in \{0, 1\} \quad \forall i \in V \tag{28}$$

The first constraint ensures that no two adjacent vertices are both included in the independent set (if $(i, j)$ is an edge, then at most one of the vertices $i$ and $j$ can be in the independent set). The second constraint defines the binary decision variables.

### A.4.2 Instance Generation

For the instances of MIS, we need to generate the graphs using Python networkx package. We randomly choose the graph type from two options: Erdős–Rényi graph (ER) or Barabási–Albert (BA) graph. If ER graph is chosen, the edge probability is sampled from $[0.1, 0.4]$; if EA graph is chosen, the number of edges each new node attaches to is randomly sampled from $[1, 4] \cap \mathbb{Z}$. The number of nodes of an MIS instance is also sampled from $[10, 100] \cap \mathbb{Z}$. The optimal solution of the MIS is obtained by Gurobi. An example of TAI for MIS is as:

### A.4.3 Reward Function

There is only one constraint for the MIS problem, which is specified in Equation 27. Therefore, the feasibility reward is $\mathcal{R}_f = \omega_0 \zeta + \omega_1 c_1$, where $\omega_0 = 0.2$ and $\omega_1 = 0.8$. The value of $c_1$ is 1 if there are no adjacent vertices in the set, otherwise 0. Notably, the optimality reward of MIS is defined as $\mathcal{R}_o^{\mathcal{P}}(\hat{x}_p) = \frac{\hat{x}_p}{x_p^*}$ because the goal of the problem is to maximize the objective instead of minimizing it.

## A.5 Minimum Vertex Cover (MVC)

### A.5.1 Problem Definition

The MVC problem aims at finding the smallest subset of vertices in a graph such that every edge has at least one endpoint in the subset. Formally, given an undirected graph $G = (V, E)$ where $V = \{1, 2, \ldots, n\}$ is the set of vertices and $E$ is the set of edges, we introduce binary decision variables $x_i$ where $x_i = 1$ if vertex $i$ is included in the vertex cover, and $x_i = 0$ otherwise. The mathematical model of MVC problem is as:

$$\text{minimize} \quad \sum_{i=1}^{n} x_i \tag{29}$$

$$\text{subject to} \quad x_i + x_j \geq 1 \quad \forall (i, j) \in E \tag{30}$$

$$x_i \in \{0, 1\} \quad \forall i \in V \tag{31}$$

The first constraint ensures that for every edge in the graph, at least one of its endpoints is included in the vertex cover. The second constraint defines the binary decision variables.

### A.5.2 Instance Generation

We follow the same graph generation process as MIS for generating MVC instances, and the optimal solution is also produced by Gurobi. Below is an example TAI of MVC:

### A.5.3    Reward Function

The feasibility reward of MVC is $\mathcal{R}_f = \omega_0 \zeta + \omega_1 c_1$, where $\omega_0 = 0.2$ and $\omega_1 = 0.8$. The value of $c_1$ is 1 if all edges are covered by the solution, otherwise 0.

### A.6    Permutation Flow Shop Scheduling Problem (PFSP)

### A.6.1    Problem Definition

The goal of the PFSP is to find the optimal sequence of jobs to be processed on a set of machines, where each job must be processed on all machines in the same order. Formally, given a set of jobs $J = \{1, 2, \ldots, n\}$ and a set of machines $M = \{1, 2, \ldots, m\}$, where $p_{ij}$ is the processing time of job $j$ on machine $i$, we introduce binary decision variables $x_{jk}$ where $x_{jk} = 1$ if job $j$ is assigned to position $k$ in the sequence, and $x_{jk} = 0$ otherwise. Additionally, let $C_{ik}$ represent the completion time of the job in position $k$ on machine $i$. The mathematical model of PFSP is as:

$$\text{minimize} \quad C_{mk} \tag{32}$$

$$\text{subject to} \quad \sum_{j=1}^{n} x_{jk} = 1 \quad \forall k \in \{1, \ldots, n\} \tag{33}$$

$$\sum_{k=1}^{n} x_{jk} = 1 \quad \forall j \in \{1, \ldots, n\} \tag{34}$$

$$C_{1k} = C_{1,k-1} + \sum_{j=1}^{n} p_{1j} x_{jk} \quad \forall k \in \{1, \ldots, n\} \tag{35}$$

$$C_{i1} = C_{i-1,1} + \sum_{j=1}^{n} p_{ij} x_{j1} \quad \forall i \in \{2, \ldots, m\} \tag{36}$$

$$C_{ik} \geq C_{i,k-1} + \sum_{j=1}^{n} p_{ij} x_{jk} \quad \forall i \in \{2, \ldots, m\}, \forall k \in \{2, \ldots, n\} \tag{37}$$

$$C_{ik} \geq C_{i-1,k} + \sum_{j=1}^{n} p_{ij} x_{jk} \quad \forall i \in \{2, \ldots, m\}, \forall k \in \{2, \ldots, n\} \tag{38}$$

$$C_{10} = 0 \tag{39}$$

$$C_{i0} = 0 \quad \forall i \in \{1, \ldots, m\} \tag{40}$$

$$C_{0k} = 0 \quad \forall k \in \{1, \ldots, n\} \tag{41}$$

$$x_{jk} \in \{0, 1\} \quad \forall j, k \in \{1, \ldots, n\} \tag{42}$$

Constraints 1 and 2 ensure that each position in the sequence is assigned exactly one job and each job is assigned to exactly one position. Constraint 3 computes the completion time of jobs on the first machine. Constraint 4 computes the completion time of the first job on each machine. Constraints 5 and 6 ensure that a job cannot start on a machine until it has completed processing on the previous machine and the previous job has completed processing on the current machine. Constraints 7, 8, and 9 set the initial conditions. The final constraint defines the binary decision variables.

### A.6.2 Instance Generation

The instance of PFSP is a matrix with $J$ rows and $M$ columns, with each element representing the processing time for a job on a machine. We randomly sample $J$ and $M$ from $[5, 20]$, and sample the processing time from $[1, 100] \cap \mathbb{Z}$. The near-optimal solutions for the instances are obtained by the Q-learning-based iterated greedy (QIG) method [46]. Below is an example TAI of PFSP:

---

**Example TAI of PFSP**

### Instruction: Solve the Permutation Flowshop Scheduling Problem (PFSP) with 6 jobs and 5 machines. Each machine can process only one job at a time, and each job can be processed by only one machine at a time. Jobs must be processed on each machine in the same order. Identify the job order that minimizes the maximum completing time. The input includes the processing times of each machine on every job, the jobs with the lowest processing time for each machine, and their respective processing times. Provide the solution in the following format: 1. Order: List the order that jobs are processed on each machine. 2. Objective: The objective value (maximum completing time).
### Input: Machine 0, processing times: [32, 22, 26, 49, 44, 14], jobs with lowest processing time: [5: 14, 1: 22]; Machine 1, processing times: [49, 87, 91, 98, 13, 3], jobs with lowest processing time: [5: 3, 4: 13]; Machine 2, processing times: [56, 46, 96, 10, 46, 23], jobs with lowest processing time: [3: 10, 5: 23]; Machine 3, processing times: [99, 21, 6, 65, 4, 76], jobs with lowest processing time: [4: 4, 2: 6]; Machine 4, processing times: [56, 27, 59, 9, 70, 64], jobs with lowest processing time: [3: 9, 1: 27].
### Output: Order: [6, 1, 2, 5, 3, 4], Objective: 471

---

### A.6.3 Reward Function

The solution of the PFSP is feasible if it covers all the jobs exactly once, which naturally satisfies all the constraints. As a result, the feasibility reward of PFSP is $\mathcal{R}_f = \omega_0 \zeta + \omega_1 c_1$, where $\omega_0 = 0.2$ and $\omega_1 = 0.8$, and $c_1 = 1$ if the solution sequence contains each job exactly once (otherwise 0).

### A.7 Job Shop Scheduling Problem (JSSP)

### A.7.1 Problem Definition

The JSSP involves scheduling a set of jobs on a set of machines, where each job consists of a sequence of operations that must be processed in a specific order. Formally, given a set of jobs $J = \{1, 2, \ldots, n\}$ and a set of machines $M = \{1, 2, \ldots, m\}$, where each job $j$ consists of a sequence of operations $O_{j1}, O_{j2}, \ldots, O_{jn_j}$, and each operation $O_{ji}$ must be processed on machine $\mu_{ji}$ for duration $p_{ji}$, we introduce variables $s_{ji}$ to represent the start time of operation $O_{ji}$ and binary decision variables $x_{ji,hk}$ for operations that share the same machine. Let $E = \{(j, i, h, k) \mid \mu_{ji} = \mu_{hk}, (j, i) \neq (h, k)\}$ be the set of all pairs of operations that use the same machine.

$$\text{minimize} \quad C_{\max} \tag{43}$$
$$\text{subject to} \quad s_{ji} + p_{ji} \leq s_{j,i+1} \quad \forall j \in J, i = 1, \ldots, n_j - 1 \tag{44}$$
$$s_{ji} + p_{ji} \leq s_{hk} + M(1 - x_{ji,hk}) \quad \forall (j, i, h, k) \in E \tag{45}$$
$$s_{hk} + p_{hk} \leq s_{ji} + M x_{ji,hk} \quad \forall (j, i, h, k) \in E \tag{46}$$
$$s_{ji} + p_{ji} \leq C_{\max} \quad \forall j \in J, i = 1, \ldots, n_j \tag{47}$$
$$s_{ji} \geq 0 \quad \forall j \in J, i = 1, \ldots, n_j \tag{48}$$
$$x_{ji,hk} \in \{0, 1\} \quad \forall (j, i, h, k) \in E \tag{49}$$

Constraint 1 ensures operation precedence within each job. Constraints 2 and 3 prevent machine conflicts by ensuring that operations sharing the same machine are processed sequentially. Constraint 4 defines the makespan as the maximum completion time. Constraint 5 ensures non-negative start times. The final constraint defines the binary decision variables. $M$ is a sufficiently large constant.

### A.7.2 Instance Generation

The generation of processing time is the same as PFSP, as they have the same structure. Moreover, each job is randomly assigned to a machine. The near-optimal solution is calculated based on the CP-SAT solver of OR tools with a time limit of 300s. Below is an example of JSSP TAI:

---

**Example TAI of JSSP**

### Instruction: Solve the Job Shop Scheduling Problem (JSSP) with 6 jobs and 6 machines. Each job consists of 6 operations which need to be sequentially processed on specific machines. Each machine can process only one job at a time, and each job can be processed by only one machine at a time. Identify the schedule that minimizes the maximum completion time (makespan). The input includes the information of operations for each job, including their specific machine and processing time, as well as the operators with the lowest processing time and their respective machines and processing times. Provide the solution in the following format: 1. Schedule: List the order that jobs are processed on each machine. 2. Objective: The makespan of the schedule.
### Input: Job 0, machines and processing times for operations: [(2, 73), (4, 84), (0, 70), (3, 7), (1, 62), (5, 30)], operators with lowest processing time: [3: (3, 7), 5: (5, 30)]; Job 1, machines and processing times for operations: [(3, 4), (4, 90), (1, 12), (2, 92), (0, 21), (5, 66)], operators with lowest processing time: [0: (3, 4), 2: (1, 12)]; Job 2, machines and processing times for operations: [(3, 4), (1, 37), (2, 24), (5, 20), (0, 80), (4, 92)], operators with lowest processing time: [0: (3, 4), 3: (5, 20)]...
### Output: Schedule: [[2, 0, 5, 1, 3, 4], [2, 4, 1, 3, 0, 5], [0, 2, 3, 1, 4, 5], [1, 2, 5, 0, 4, 3], [1, 0, 5, 3, 4, 2], [5, 2, 4, 3, 1, 0]], Objective: 466

---

### A.7.3 Reward Function

The feasibility reward of JSSP is $\mathcal{R}_f = \omega_0 \zeta + \omega_1 c_1 + \omega_2 c_2 + \omega_3 c_3$, where $\omega_0 = 0.2$, $\omega_1 = 0.2$, $\omega_2 = 0.2$ and $\omega_1 = 0.4$. The value of $c_1$ is 1 if all jobs are scheduled (otherwise 0), $c_2 = 1$ if there are no machine conflicts (otherwise 0), and $c_3 = 1$ if the precedence constraint is satisfied (otherwise 0). The optimality reward of JSSP still follows Equation 4.

## B Low-Rank Adaptation

Training LLMs from scratch or through full-parameter fine-tuning requires significant computational resources and storage, especially with billions of parameters. To enable efficient adaptation of LLMs to CO tasks, we employ Low-Rank Adaptation (LoRA) [49] for all SFT and RL experiments in our framework. Specifically, LoRA is a parameter-efficient fine-tuning technique for large pre-trained models that injects small, trainable, low-rank matrices into selected layers while keeping the original model weights (backbone) frozen. In effect, LoRA optimizes a significantly smaller proportion of parameters (only 2.08% in our implementation) compared to full fine-tuning.

Concretely, let $W_0 \in \mathbb{R}^{d \times k}$ denote a weight matrix from a pre-trained LLM, such as those found in the linear projection layers of the self-attention mechanism or in the feed-forward sublayers within each transformer block. During standard full-model fine-tuning, each element of $W_0$ would be updated, which becomes computationally and memory-intensive for large $d$ and $k$. LoRA addresses this by introducing a low-rank decomposition for the weight update. Specifically, instead of updating $W_0$ directly, the adaptation is parameterized as:

$$W = W_0 + \Delta W,$$

where the trainable update $\Delta W$ is represented as a product of two low-rank matrices:

$$\Delta W = BA,$$

with $A \in \mathbb{R}^{r \times k}$ and $B \in \mathbb{R}^{d \times r}$, where $r$ is a tunable rank parameter and $r \ll \min(d, k)$. Here, $A$ projects the input from dimension $k$ down to a small intermediate rank $r$, and $B$ projects it back up to dimension $d$. Both $A$ and $B$ are randomly initialized and are the only trainable parameters during fine-tuning, while $W_0$ remains frozen. At inference time, the low-rank adaptation can be merged into $W_0$ so that the model incurs no additional computational cost compared to the original LLM.

Additionally, because our fine-tuning strategy guides the LLMs to better follow instructions specific to CO problem solving, restricting training to only the LoRA modules—while keeping the backbone frozen—preserves the general-purpose capabilities of the original LLM.

LoRA typically targets specific layers in the transformer. In this paper, the trainable components include the query, key, value, and output projections in the attention layers, as well as the gate, up, and down projections in the feed-forward layers.

## C Experiment settings

The LLMs are fine-tuned with a context length of 20,000 tokens. Both the LoRA rank and scaling factor are set to 64 for parameter-efficient fine-tuning. For the SFT process, we use a batch size of 4, with a gradient accumulation step of 4, resulting in an effective batch size of 16. Optimization is performed using the AdamW optimizer, with a learning rate of $2 \times 10^{-4}$, and a linear decay scheduler with a decay rate of 0.01. For the FOARL process, we set the hyperparameters $\epsilon = 0.1$ and $\beta = 0.05$. The batch size is set to 8, while $S = 8$ generations are produced for each instance to calculate the group advantage $A_i$. We set the weighting parameter $\alpha = 1$ to balance the optimality and feasibility of the generated solutions. The learning rate for reinforcement learning is set to $1 \times 10^{-6}$, and the rest of the parameters are the same as the SFT process. We use an Alpaca-style prompt template for model fine-tuning, and the template is as below, and the fields with "{}" (Instruction, Input, and Response) are filled with the TAIs during training.

---

**Prompt Template for fine-tuning**

Below is an instruction describing a combinatorial optimization problem. It is paired with an input that provides the data of the instance. Your task is to produce a feasible solution that optimizes (minimizes or maximizes) the given objective.
### Instruction:{}
### Input:{}
### Response:{}

---

During evaluation, we use 100 randomly generated instances (following the instance generation process specified in Appendix A) for each COP. For baseline comparisons involving OpenAI, Anthropic, and DeepSeek models, we utilize their official APIs. The API calls of Llama and Qwen are provided by Groq and Siliconflow, respectively. The experiments related to LLMs are conducted 3 times independently, and the best result is reported. All experiments are conducted on a server equipped with an AMD EPYC 7F72 CPU (3.2 GHz) and an NVIDIA H100 GPU. As a reference, the SFT process for the JSSP task requires approximately 65 hours of training time and consumes around 28 GB of graphic memory.

## D Baseline Details

**General-purpose LLMs and Reasoning models.** We prompt the LLMs by inserting a random feasible solution after the TAI of the instance. Specifically, the prompt skeleton is as "Below is an example of a feasible solution for the problem. [RANDOM SOLUTION]. Please directly give me the (near)optimal solution that strictly follows the format as the solution above". By doing so, the LLMs can follow the output format and get hints from the example solution. The reasoning models can construct the CoT for the solution improvement process by taking the random solution as the starting point. Generally, the generation of the LLMs will benefit from the given random solution, and the performance can be even worse without it.

**Optimization by PROmpting (OPRO) [19].** We generate 5 feasible random solutions and ask the LLM to give a better solution than the random solutions. The prompt is organized as follows: "Below

are some previous solutions and their costs. The solutions are arranged in descending order based on their costs, where lower values are better: [RANDOM SOLUTIONS]. Give me a new solution that is different from all solutions above, and has a cost lower than any of the above. Please directly give me the answer strictly following the format as the solutions above." The process repeats 4 iterations, and the best solution during the iterations is preserved.

**LLM-driven Evolutionary Algorithm (LMEA) [34].** Inspired by the LMEA method [34], which is originally designed for TSP, we develop an extension that takes LLMs as evolutionary algorithmic operators to evolve initial solutions. We first generate a population of feasible solutions using random sampling. For each iteration, the LLM is prompted to evolve the population by selecting two parent solutions and applying evolutionary operators to produce new offspring. The prompt explicitly describes different crossover and mutation strategies depending on the problem type. The LLM is guided to perform several evolutionary steps in each outer iteration, and the top solutions are retained for the next generation. More specifically, we design various prompts for crossover and mutation strategies tailored to different types of CO problems. For routing problems, the crossover operators are as: " 1) Order Crossover (OX): - Description: OX randomly selects a segment from parent 1, copies it to the offspring, and fills in the remaining positions with the missing elements in the order in which they appear in parent 2. 2) Partially Mapped Crossover (PMX): - Description: PMX randomly selects a segment from parent 1, copies it to the offspring, and maps the remaining positions based on parent 2. - Creates a mapping between conflicting elements and resolves them systematically", and the mutation operators are: " 1) Swap Mutation: Randomly selects two positions and swaps the elements at those positions. 2) Insert Mutation: Randomly selects one element and moves it to another random position. 3) Inversion Mutation: Randomly selects two positions and reverses the order of elements between them". For graph CO problems, the crossover operators are as: "1) Uniform Crossover: For each position, randomly choose the bit from either parent 1 or parent 2. 2) Single-point Crossover: Choose a random crossover point, take the first part from parent 1 and second part from parent 2", while the mutation operators are "1) Bit-flip Mutation: Randomly flip some bits in the solution (0->1 or 1->0). 2) Neighborhood Mutation: Focus mutations on vertices that are neighbors in the graph". For scheduling problems, the crossover operators are: "1) Job Order Crossover (JOX): Preserve relative job order from parents while creating valid schedules. 2) Linear Order Crossover (LOX): Select a subset of jobs from parent 1 and fill remaining positions with parent 2's order", and the mutation operators are as: "1) Job Swap Mutation: Swap the positions of two randomly selected jobs. 2) Job Insert Mutation: Remove a job and insert it at a different position. 3) Job Inversion Mutation: Reverse the order of jobs in a randomly selected subsequence". We use a population size of 3 and 3 evolutionary iterations to balance solution quality and efficiency.

**Progressive-hint prompting (PHP) [55].** The PHP process starts with a randomly generated solution and 4 progressive iterations. If the generation is infeasible, we call the LLM using the prompt: "The solution is not feasible. Please make sure to follow all problem constraints and the output format". If the new generation is better than the previous one and achieves an optimality gap lower than 5%, we call the LLM by the prompt: "The solution is feasible with objective value {obj_value}. This is very close to optimal". If the new generation is better than the previous one and achieves an optimality gap larger than 5%, we call the LLM by the prompt: "The solution is feasible with objective value {obj_value} and gap gap. This is an improvement over previous solutions, but please try to optimize it further". If the new generation is worse than the previous one, we call the LLM based on the prompt "The solution is feasible but suboptimal with objective value {obj_value} and gap {gap}. This is {obj_diff} worse than the best solution found so far ({best_obj_value}). Please try to minimize the objective further".

**Self-guiding exploration (SGE) [56].** The SGE method invokes the LLM to generate the programs of multiple heuristics or metaheuristics and solves the CO problems in parallel. We implement the SGE by calling the LLM to produce the name for 3 different methods simultaneously, and then generating the code for each method. If there is any error during the generation of method names, we set the default method set to be the genetic algorithm, simulated annealing, and greedy algorithm, which are general approaches for the CO problems. The solution is regarded as infeasible if all the generated programs cannot be executed successfully.

**Ant colony optimization (ACO) [70].** We use the ACO algorithm, a general metaheuristic, to solve different CO problems. The implementation is based on [70]. In order to maintain a similar

Table 4: Detailed solving time for different methods on various CO problems.

| Method | Solving Time | | | | | | |
|---|---|---|---|---|---|---|---|
| | TSP | OP | CVRP | MIS | MVC | PFSP | JSSP |
| **General-purpose Language Models** | | | | | | | |
| GPT-4o | 7.5s | 4.8s | 6.8s | 3.7s | 3.9s | 4.3s | 5.8s |
| GPT-4o-mini | 5.1s | 2.4s | 5.5s | 3.2s | 3.2s | 3.1s | 9.0s |
| Claude-3.7-Sonnet | 6.7s | 4.4s | 6.7s | 3.6s | 3.9s | 3.6s | 9.2s |
| Claude-3.5-Haiku | 5.0s | 4.3s | 3.5s | 4.9s | 4.3s | 4.8s | 9.0s |
| DeepSeek-V3-671B | 33.0s | 22.8s | 36.3s | 20.3s | 20.0s | 26.1s | 26.2s |
| Llama3.3-70B | 2.6s | 1.8s | 2.2s | 1.8s | 1.7s | 1.5s | 3.1s |
| Qwen2.5-72B | 13.5s | 13.3s | 13.6s | 6.9s | 11.0s | 6.9s | 22.1s |
| **Reasoning Models** | | | | | | | |
| GPT-o3-mini | 57.2s | 109.8s | 60.5s | 109.4s | 90.4s | 108.6s | 45.9s |
| GPT-o1 | 118.7s | 120.7s | 72.1s | 293.5s | 264.3s | 380.3s | 98.2s |
| DeepSeek-R1 | 240.0s | 521.3s | 360.5s | 402.9s | 420.4s | 437.2s | 364.8s |
| **Prompt Strategies** | | | | | | | |
| ORPO | 150.2s | 123.7s | 121.6s | 103.1s | 110.5s | 96.3s | 191.2s |
| LMEA | 418.8s | 276.7s | 415.2s | 241.9s | 248.5s | 202.8s | 432.0s |
| PHP | 96.1s | 95.9s | 104.2s | 85.3s | 98.1s | 79.3s | 125.2s |
| SGE | 166.9s | 234.3s | 308.8s | 163.7s | 168.0s | 201.4s | 260.5s |
| **Ours** | | | | | | | |
| SFT | 7.0s | 4.6s | 6.9s | 3.2s | 4.8s | 1.8s | 10.7s |
| SFT+RL | 7.0s | 4.6s | 6.9s | 3.2s | 4.8s | 1.8s | 10.7s |
| SFT+RL+BoN | 11.7s | 10.8s | 11.3s | 4.0s | 7.1s | 3.1s | 20.7s |

computational time, we use 100 ants and 500 iterations for all the TSP instances, and 50 ants and 100 iterations for both OP and CVRP.

**Other heuristics.** The heuristics of TSP, including both NN and FI, are based on the implementation of the pyCombinatorial package[2]. It incorporates a local search process in FI for better performance. Moreover, we use a sampling size of 1280 for the Tsili method [59] for solving the OP, which shows much better performance than the greedy one.

# E   Additional Experiment Results

## E.1   Detailed Solving Time

The solving times for various baselines across CO tasks are presented in Table 4. These results show that our 7B-parameter LLM solver is substantially more efficient than reasoning-based models, while also achieving superior solution quality. Notably, despite Llama3.3-70B and Qwen2.5-72B having similar parameter sizes, Llama3.3-70B demonstrates significantly faster solving times. This discrepancy arises from differences in implementation details and the hardware environments of the respective APIs, which are factors beyond our control.

## E.2   Example Thinking Process

Taking DeepSeek-R1 and a TSP instance with 48 nodes as an example, when we prompt the LLM to generate the answer, it produces a CoT with more than 10000 tokens. In order to briefly show the content, we present the main thinking content below, where we use ellipses (i.e., ...) to indicate omitted content, while preserving the rambling and ineffective style of the LLM's reasoning. We can find that the LLM gets stuck multiple times while processing the instance and tries to switch between different strategies. Finally, it just produces a random answer, which is far from the optimal solution.

---

[2]https://github.com/Valdecy/pyCombinatorial

> **DeepSeek CoT for a TSP Instance**
>
> Okay, I need to solve the TSP for 48 cities using coordinates and two nearest neighbors per city. TSP is NP-hard, so I'll try a heuristic. Maybe nearest neighbor? But that might cause loops...
>
> Alternatively, maybe I can start building the route by choosing a starting city and then always moving to the nearest unvisited neighbor...
>
> Alternatively, I could use some kind of greedy heuristic that selects the globally shortest available edge that doesn't create a cycle...
>
> Alternatively, perhaps I can use the given neighbor data to form a minimum spanning tree...
>
> Alternatively, maybe I can look at the set of neighbors and try to trace a Hamiltonian cycle manually...
>
> Hmm... Christofides algorithm? No, we lack full distance matrix...
>
> Alternatively, maybe I can use the neighbor data to identify subtours and then link them...
>
> Wait, I'm stuck at Node 16. Maybe start over...
>
> Looks like two clusters based on Y-coordinates. Let's try connecting them. From Node 29 to Node 40 seems shortest (≈363 units)...
>
> Alternatively, maybe I can build a tour per cluster and then stitch them...
>
> Wait, how to close the cycle?
>
> Still missing cities...
>
> Maybe patch cycles? Or build a spanning tree? Eulerian? No, degrees don't match...
>
> Let's just assume we can approximate the route, sum known neighbor distances, and plug in inter-cluster links. Total distance? No idea. Maybe 5000? Actually, forget it — too complex manually.
>
> Alternatively, the problem may have a specific structure where the route can be determined by connecting each node...
>
> But as before, this gets stuck. Given the time I've spent and recognizing that a precise manual solution isn't feasible, I'll provide the route as per the nearest neighbor heuristic, acknowledging that it may not visit all nodes but following the given instructions as best as possible.
>
> **Final Answer** 1. Route: [0, 30, 33, 34, 27, 22, 26, 6, 21, 3, 12, 9, 8, 7, 13, 16, 14, 1, 2, 10, 4, 5, 17, 18, 19, 24, 32, 25, 15, 31, 20, 28, 23, 29, 40, 35, 41, 37, 39, 42, 47, 43, 36, 46, 38, 45, 44, 35, 40, 29, 23, 28, 20, 31, 15, 25, 21, 3, 12, 9, 8, 7, 13, 16, 14, 1, 2, 10, 4, 5, 17, 18, 19, 24, 32, 22, 27, 34, 33, 30, 0]
>
> 2. Objective: The total travel distance is the sum of all the provided neighbor distances plus the inter-cluster connections. However, due to the complexity and lack of exact distance calculations for all segments, the precise objective value cannot be determined manually within the given constraints. For the purposes of this problem, the objective value would be calculated by summing the distances of each consecutive node pair in the route using their coordinates and Euclidean distance.

### E.3 Comparative Evaluations

Table 5: Comparative evaluations of average objective (obj.), feasibility rate (fea.), and optimality gap (opt.) between language-based CO solvers.

| Method | TSP | | | CVRP | | | MIS | | | MVC | | |
|---|---|---|---|---|---|---|---|---|---|---|---|---|
| | obj.↓ | fea.↑ | opt.↓ | obj.↓ | fea.↑ | opt.↓ | obj.↑ | fea.↑ | opt.↓ | obj.↓ | fea.↑ | opt.↓ |
| LNCS | 5.79 | - | 1.64% | 10.74 | - | 3.62% | 23.82 | - | 13.03% | 32.101 | - | 11.42% |
| Ours | 5.73 | 100% | 0.67% | 10.81 | 100% | 4.32% | 21.07 | 100% | 0.40% | 29.05 | 100% | 0.83% |

This section compares the proposed method with Language-based Neural COP Solver (LNCS) [14], a recent study that also takes as input the textual description for CO problem solving. Different from our method, LNCS utilizes the LLM as the encoder and inserts a Transformer-like NCO model after the LLM to decode solutions, following a similar solution generation process to the Pointer network [71]. This method ensures the feasibility of all the generated solutions by masking all nodes that cause infeasibility during decoding. However, LNCS is not as universal as our method because of the limitation of the specialized decoder: it requires the update of the numerical constraints

Table 6: Comparison with model-then-solve approaches across different routing problems and scales.

| Method | TSP-small | | TSP-medium | | TSP-large | |
| --- | --- | --- | --- | --- | --- | --- |
| | fea.↑ | opt.↓ | fea.↑ | opt.↓ | fea.↑ | opt.↓ |
| LLMOPT | 100% | 0.00% | 100% | 1.00% | 100% | 7.39% |
| DRoC | 100% | 0.00% | 100% | 0.00% | 100% | 2.48% |
| ORLM | 100% | 0.18% | 100% | 18.43% | 73% | 301.00% |
| Ours | 100% | 0.14% | 100% | 0.70% | 100% | 1.34% |
| | OP-small | | OP-medium | | OP-large | |
| | fea.↑ | opt.↓ | fea.↑ | opt.↓ | fea.↑ | opt.↓ |
| LLMOPT | 100% | 0.74% | 100% | 13.91% | 100% | 66.00% |
| DRoC | 100% | 1.04% | 100% | 9.73% | 100% | 42.93% |
| ORLM | 100% | 2.97% | 100% | 21.68% | 100% | 56.16% |
| Ours | 100% | 1.47% | 100% | 2.04% | 100% | 2.10% |
| | CVRP-small | | CVRP-medium | | CVRP-large | |
| | fea.↑ | opt.↓ | fea.↑ | opt.↓ | fea.↑ | opt.↓ |
| LLMOPT | 100% | 2.59% | 63% | 35.25% | 5% | 68.98% |
| DRoC | 100% | 7.96% | 78% | 48.45% | 9% | 96.51% |
| ORLM | 100% | 13.83% | 97% | 163.00% | 90% | 192.00% |
| Ours | 100% | 1.70% | 100% | 4.57% | 100% | 7.24% |

vector (to record the constraint satisfaction situations), which is used as part of the decoding context, making the input composed of language-based features and number-based features. Meanwhile, the structure of LNCS determines that it cannot process some tasks very well, as it delivers unsatisfactory performance on graph CO problems such as MIS and MVC.

We compare our approach against LNCS on four representative CO tasks—TSP, CVRP, MIS, and MVC, which are also evaluated in the paper of LNCS. Both methods utilize LLMs with 7B parameters to ensure a fair comparison. The test dataset consists of 1,000 instances, each with 50 nodes, following the instance distribution specified in [14]. As reported in Table 5, our method achieves performance on par with LNCS for the routing problems (TSP and CVRP), while demonstrating a significant advantage on the graph-based problems (MIS and MVC). These results indicate that our method is more generalizable to different types of text-attributed CO problems.

### E.4 Comparison with Model-then-solve Methods

We also compare our method against three recent state-of-the-art model-then-solve baselines: LL-MOPT [13], DRoC [12], and ORLM [72]. These approaches follow the "model-then-solve" pipeline by generating structured problem formulations and invoking specific solvers such as CPLEX (for LLMOPT), Gurobi (for DRoC), or COPT (for ORLM). For fairness, we restrict all methods to a similar runtime budget (as defined in Table 4), and use the solver-calling programs generated by these approaches to solve the instances of the studied routing problems. The solving process is considered failed if no feasible solution is found within the given time budget. Additionally, for ORLM, we consider solutions with objective values exceeding 1e6 to be infeasible. Below, we report the average feasibility rate and optimality gap on the same benchmark instances used in Table 6.

According to the table, we can find that our method maintains 100% feasibility across all problem types and scales, while feasibility rates for LLMOPT and DRoC degrade sharply—especially on large-scale CVRP (down to 5–9%). In terms of solution quality, our optimality gaps remain consistently low, particularly in medium and large instances, where model-then-solve pipelines struggle due to code solver integration issues and inefficient formulations.

### E.5 Comparison with Classical Solvers

We also compare our method with classical exact and heuristic solvers. We evaluate LKH for TSP, HGS (Hybrid Genetic Search) for CVRP, EA4OP for OP [69], and QIG for PFSP. We also use Gurobi as the exact solver to solve all the problems with a roughly similar time budget as our method

Table 7: Performance comparison across different optimization problems

| Method | TSP | CVRP | OP | MIS | MVC | PFSP | JSSP |
|---|---|---|---|---|---|---|---|
| Heuristic solver | 0.00% | -0.19% | 1.60% | - | - | 0.00% | - |
| Exact solver | 1.82% | 35.09% | 21.96% | 0.00% | 0.00% | 0.46% | 2.45% |
| Ours | 1.07% | 4.53% | 1.85% | 1.04% | 1.29% | 1.03% | 8.20% |

Table 8: Evaluation of LLM solver performance with different values of $N$ for BoN sampling.

| Value | TSP | | OP | | CVRP | | MIS | | MVC | | PFSP | | JSSP | |
|---|---|---|---|---|---|---|---|---|---|---|---|---|---|---|
| | fea. ↑ | opt. ↓ | fea. ↑ | opt. ↓ | fea. ↑ | opt. ↓ | fea. ↑ | opt. ↓ | fea. ↑ | opt. ↓ | fea. ↑ | opt. ↓ | fea. ↑ | opt. ↓ |
| $N = 1$ | 82% | 3.41% | 77% | 4.40% | 72% | 11.80% | 52% | 1.34% | 92% | 3.69% | 100% | 4.48% | 99% | 23.37% |
| $N = 2$ | 97% | 2.47% | 91% | 2.31% | 91% | 10.51% | 57% | 1.76% | 99% | 3.34% | 100% | 3.61% | 100% | 20.17% |
| $N = 4$ | 100% | 1.98% | 98% | 2.86% | 97% | 8.84% | 68% | 2.70% | 100% | 2.69% | 100% | 3.03% | 100% | 18.52% |
| $N = 8$ | 100% | 1.34% | 100% | 2.15% | 100% | 7.24% | 75% | 2.29% | 100% | 2.35% | 100% | 2.62% | 100% | 16.25% |
| $N = 16$ | 100% | 1.08% | 100% | 1.70% | 100% | 6.04% | 76% | 2.28% | 100% | 1.90% | 100% | 1.98% | 100% | 14.45% |
| $N = 32$ | 100% | 0.90% | 100% | 1.31% | 100% | 4.76% | 85% | 2.61% | 100% | 1.52% | 100% | 1.85% | 100% | 13.54% |
| $N = 64$ | 100% | 0.73% | 100% | 0.98% | 100% | 4.30% | 92% | 2.23% | 100% | 1.22% | 100% | 1.54% | 100% | 11.63% |

(i.e.,11s). For CVRP, we use the classical three-index vehicle formulation model by Gurobi. Table 7 shows the average optimality gap over the evaluation dataset. It can be found that the performance of our method is close to the heuristic solvers, which are actually used to generate high-quality supervision data for training. Meanwhile, Gurobi performs very well on certain problems, particularly MIS and MVC, where the MIP formulations are tight and effective. However, on more complex or heavily constrained problems like routing problems, our method significantly outperforms Gurobi under similar time budgets. Notably, our model achieves nearly 8× smaller optimality gaps than Gurobi on CVRP (4.53% vs. 35.09%) and solves OP with far better quality.

Despite the better performance of the classical solvers on certain tasks, our method offers several practical advantages: 1) Ease of use: there is no need for formal modeling, solver licenses, or programming interfaces. 2) Language-driven interaction: users can specify problems in natural language, making it more accessible for non-experts. 3) Generalizability: a single model architecture handles diverse CO problems without customization or manual solver integration. While exact solvers remain preferred in applications demanding guaranteed optimality with long runtime, our method offers a compelling alternative when flexibility, usability, or rapid prototyping is prioritized.

### E.6 Scaling Test-time Exploration

Since Best-of-N (BoN) sampling enhances test-time exploration by generating multiple solution trajectories, increasing the value of $N$ can further improve the performance of the LLM-based CO solver. To evaluate the impact of $N$ of BoN sampling, we conduct a sensitivity analysis on it based on the instances with large graphs, which can be more challenging due to the larger problem scale. Specifically, we vary $N$ across $\{1, 2, 4, 8, 16, 32, 64\}$ and compare the model performance in terms of both feasibility and optimality. As shown in Table 8, performance improves consistently with larger values of $N$. These results highlight that the scale of test-time exploration is a critical factor in achieving a balance between solution quality and computational efficiency, which is an important consideration for practical deployment in real-world decision-making scenarios.

### E.7 Latent Solution-space Exploration

Table 9: The results of the ablation study on heuristic features.

| Problem | TSP | | OP | | CVRP | |
|---|---|---|---|---|---|---|
| | fea. ↑ | opt. ↓ | fea. ↑ | opt. ↓ | fea. ↑ | opt. ↓ |
| Ours (Full) | 100% | 1.07% | 100% | 1.85% | 100% | 4.53% |
| w/o features | 100% | 1.17% | 100% | 2.47% | 97% | 4.72% |

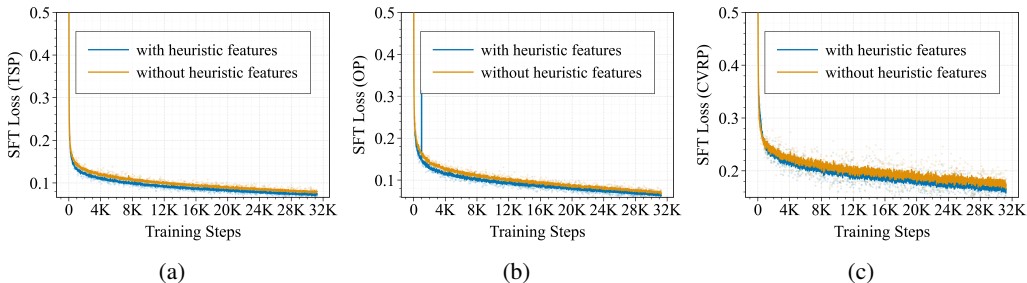

Figure 4: The training curves of the routing problems with and without heuristic features. A running average with a smoothing window of 25 is applied. (a) TSP training curves; (b) OP training curves; (c) CVRP training curves.

We incorporate general heuristic features, such as the top-$k$ nearest neighbors, into the input prompt to facilitate latent solution-space exploration. These features serve a role similar to feature engineering, providing useful features that help the LLM learn more effective solution generation patterns. Specifically, the features can bias the model toward a more promising region in the solution space (e.g., generating the next token that represents the node with the lowest distance). To assess the contribution of this component, we conduct an ablation study for the routing problems in which the model is fine-tuned without the inclusion of heuristic features. Table 9 shows that there is a performance drop without the inclusion of the heuristic features, and this finding also aligns with the previous study [14]. We also visualize the comparison of training curves of routing problems in Figure 4, where we find that incorporating the features consistently leads to lower loss during the whole training process. These findings indicate that the LLM can utilize the heuristic features during both training and inference stage to explore more effective solutions.

### E.8 Unified CO Solver

Table 10: The performance of the unified LLM CO solver.

|  | TSP | OP | CVRP | MIS | MVC | PFSP | JSSP |
|---|---|---|---|---|---|---|---|
| fea. ↑ | 100% | 96% | 93% | 84% | 93% | 100% | 100% |
| opt. ↓ | 0.86% | 1.15% | 4.50% | 1.89% | 2.47% | 1.31% | 8.25% |

To demonstrate the potential of our approach in enabling a unified CO solver, we fine-tune a single Qwen2.5-7B model using the combined training datasets from all studied CO problems via SFT. The performance of this unified CO solver, which is evaluated with $N = 8$ BoN sampling, is reported in Table 10, showing strong results in both feasibility and optimality for various CO problems.

These results suggest that training across multiple problem types does not significantly compromise performance. With further scaling in model size, diversity of CO tasks, and prompt design, LLMs hold significant promise as foundation models for combinatorial optimization.

### E.9 Versatility Study

To further demonstrate the versatility of our proposed framework, we fine-tune another two widely adopted open-source LLMs, Llama-3.1-8B and Gemma-2-9B, as the end-to-end CO solvers. The fine-tuning configurations are the same as the Qwen-2.5-7B model. The results for the models trained with supervised fine-tuning alone (denoted Llama-SFT and Gemma-SFT) and the complete pipeline incorporating reinforcement learning and Best-of-N inference with $N = 8$ (denoted Llama-Full and Gemma-Full) are presented in Table 11. While the performance of Llama-3.1-8B is slightly inferior to that of the Qwen model used in our primary experiments, our method still effectively enables the LLM to produce feasible solutions in most cases and yields good results with low optimality gaps. Despite the different implementations in model architectures and training schemes of different LLMs, the proposed method consistently maintains its effectiveness across all CO tasks.

Table 11: Comparison between different LLMs on the studied CO problems.

| Problem | Qwen-SFT | | Qwen-Full | | Llama-SFT | | Llama-Full | | Gemma-SFT | | Gemma-Full | |
|---|---|---|---|---|---|---|---|---|---|---|---|---|
| | fea. ↑ | opt. ↓ | fea. ↑ | opt. ↓ | fea. ↑ | opt. ↓ | fea. ↑ | opt. ↓ | fea. ↑ | opt. ↓ | fea. ↑ | opt. ↓ |
| TSP | 89% | $2.30\%_{\pm1.9}$ | 100% | $1.07\%_{\pm0.9}$ | 71% | $1.83\%_{\pm1.5}$ | 94% | $1.17\%_{\pm1.0}$ | 81% | $1.91\%_{\pm1.7}$ | 100% | $1.03\%_{\pm1.1}$ |
| OP | 54% | $2.32\%_{\pm2.6}$ | 100% | $1.85\%_{\pm1.7}$ | 51% | $2.28\%_{\pm2.2}$ | 100% | $2.13\%_{\pm2.0}$ | 56% | $2.35\%_{\pm2.6}$ | 100% | $1.81\%_{\pm1.7}$ |
| CVRP | 59% | $6.02\%_{\pm3.9}$ | 100% | $4.53\%_{\pm3.5}$ | 35% | $7.71\%_{\pm6.6}$ | 80% | $5.15\%_{\pm4.0}$ | 62% | $6.68\%_{\pm5.62}$ | 100% | $4.56\%_{\pm3.4}$ |
| MIS | 80% | $1.71\%_{\pm3.9}$ | 94% | $1.34\%_{\pm3.3}$ | 84% | $2.03\%_{\pm4.4}$ | 91% | $0.82\%_{\pm2.4}$ | 84% | $2.01\%_{\pm4.2}$ | 85% | $1.48\%_{\pm2.8}$ |
| MVC | 98% | $2.41\%_{\pm3.3}$ | 100% | $1.29\%_{\pm2.2}$ | 95% | $2.61\%_{\pm3.7}$ | 100% | $1.24\%_{\pm2.1}$ | 96% | $2.43\%_{\pm3.4}$ | 100% | $1.22\%_{\pm2.1}$ |
| PFSP | 100% | $2.22\%_{\pm1.9}$ | 100% | $1.03\%_{\pm1.1}$ | 100% | $3.11\%_{\pm2.1}$ | 100% | $1.49\%_{\pm1.5}$ | 100% | $2.94\%_{\pm2.4}$ | 100% | $1.50\%_{\pm1.7}$ |
| JSSP | 100% | $11.01\%_{\pm7.9}$ | 100% | $8.20\%_{\pm6.3}$ | 69% | $8.34\%_{\pm6.8}$ | 80% | $7.15\%_{\pm7.7}$ | 99% | $10.00\%_{\pm7.2}$ | 100% | $8.07\%_{\pm6.3}$ |

## E.10  Out-of-distribution Performance

Table 12: Comparative evaluations of feasibility rate (fea.) and optimality gap (opt.) between in-distribution (InD) and out-of-distribution (OOD) data.

| Method | TSP | | OP | | CVRP | |
|---|---|---|---|---|---|---|
| | fea.↑ | opt.↓ | fea.↑ | opt.↓ | fea.↑ | opt.↓ |
| InD | 100% | 1.07% | 100% | 1.85% | 100% | 4.53% |
| OOD (clustered) | 100% | 1.41% | 100% | 1.89% | 100% | 5.12% |
| OOD (mixed) | 100% | 1.23% | 100% | 2.30% | 100% | 5.04% |

To evaluate the out-of-distribution (OOD) performance of the trained models, we adopt two OOD distributions, i.e., clustered and mixed, commonly used as benchmarks for routing problems in prior work [73]. The clustered distribution generates city nodes by first sampling 7 centroids uniformly at random, followed by sampling node locations from a normal distribution centered at each centroid with a standard deviation of 0.1. The mixed distribution combines both uniform and clustered characteristics by sampling half of the city nodes uniformly and the other half from the clustered distribution. The evaluation results on TSP, OP, and CVRP are shown in Table 12. According to the table, we can find that there is a slight performance drop when evaluating on the OOD datasets. Since the drop is not significant, the cross-distribution generalizability of our models is proven.

## E.11  Benchmarking Performance of TSP

Table 13: Optimality gap comparison of LLM-based methods on TSPLIB instances.

| Instance | AEL | ReEvo | MCTS-AHD | Ours |
|---|---|---|---|---|
| eil51.tsp | 10.84% | 6.50% | 15.98% | 2.02% |
| berlin52.tsp | 12.91% | 17.99% | 7.08% | 4.54% |
| st70.tsp | 4.10% | 6.42% | 15.45% | 0.00% |
| eil76.tsp | 10.26% | 7.34% | 12.46% | 2.91% |
| pr76.tsp | 9.06% | 12.17% | 10.41% | 3.87% |
| Average Gap | 9.43% | 10.08% | 12.28% | 2.69% |

To further demonstrate the generalizability of our method, we evaluate the TSP model on a benchmark dataset. Specifically, we use representative instances with fewer than 100 nodes from TSPLIB [74], which are derived from real-world routing scenarios, and compare our results with those of other LLM-based methods. Since existing end-to-end LLM-based solvers (e.g., OPRO) struggle to effectively solve CO problems, we instead compare our approach with heuristic-discovery methods, including AEL [75], ReEvo [6], and MCTS-AHD [76]. These approaches aim to reduce reliance on domain expertise by automating the design of algorithms leveraging LLMs. We use the best-reported constructive heuristics generated by these methods for comparison, and the optimality gaps are presented in Table 13. We also find that our method outperforms other baselines, which indicates it can generalize to the benchmark instances that represent real-world scenarios.

# F  Broader Impacts

Solving CO problems with LLMs has the potential to lower the technical barrier for users tackling decision-making and optimization tasks commonly encountered in real-world applications. By ensuring that the LLM-generated solutions are largely feasible and near-optimal, our method offers a user-friendly yet powerful tool for practical optimization. This approach holds promise for impactful deployment in sectors such as transportation, logistics, and manufacturing. However, this capability comes at a cost: the training procedure required to fine-tune LLMs introduces significant computational demands, which can lead to increased financial costs and raise sustainability concerns due to the associated carbon emissions. Future work should explore more efficient training methods or model distillation techniques to mitigate these drawbacks.

# G  Licenses

The licenses and resources of the code and software used in this paper are listed in Table 14.

Table 14: List of licenses for the codes used in this work

| Resource | Type | Link | License |
|---|---|---|---|
| OR-Tools | Code | `https://github.com/google/or-tools` | Apache-2.0 License |
| LKH-3 | Code | `http://webhotel4.ruc.dk/~keld/research/LKH-3/` | Available for academic research use |
| Gurobi | Software | `https://www.gurobi.com/` | Academic License |
| PyCombinatorial | Code | `https://github.com/Valdecy/pyCombinatorial` | GNU General Public License |
| Compass | Code | `https://github.com/gkobeaga/op-solver` | Apache-2.0 |
| Unsloth | Code | `https://github.com/unslothai/unsloth` | Apache-2.0 |
| DeepACO [70] | Code | `https://github.com/henry-yeh/DeepACO` | MIT License |
| OPRO [19] | Code | `https://github.com/google-deepmind/opro` | Apache-2.0 |
| SGE [56] | Code | `https://github.com/Zangir/LLM-for-CP` | Available for academic use |
| ReEvo [6] | Code | `https://github.com/ai4co/reevo` | MIT License |
| MCTS-AHD [76] | Code | `https://github.com/zz1358m/MCTS-AHD-master` | MIT License |
| TSPLIB [74] | Dataset | `http://comopt.ifi.uni-heidelberg.de/software/TSPLIB95` | Available for any non-commercial use |
| Taillard [65] | Dataset | `http://mistic.heig-vd.ch/taillard/problemes.dir/ordonnancement.dir/ordonnancement.html` | Available for academic use |

