# OpenReview forum: "Large Language Models as End-to-end Combinatorial Optimization Solvers"
_NeurIPS.cc/2025/Conference — NeurIPS 2025 poster_

### Official Review · Reviewer_u6T1 · 2025-06-04

**Clarity:** 4
**Significance:** 3
**Originality:** 3
**Rating:** 5
**Confidence:** 3

**Summary:**

The authors leverage supervised fine-tuning (SFT) and reinforcement learning (RL) techniques to post-train a Qwen2.5-7B model for end-to-end combinatorial optimization (CO).
Problem instances with best known solutions are converted into text descriptions using the TAI tool, and the Qwen model is fine-tuned to generate solutions directly from these textual inputs. The RL phase incorporates feasibility and optimality rewards within the GPRO framework. Empirical results demonstrate surprisingly strong performance compared to other LLMs and heuristic baselines.

**Questions:**

1. Since the paper does not emphasize the LLM’s reasoning process, what explains its strong performance? Have you compared against supervised GNN-based methods to provide further context?

2.  How well does the method generalize to unseen problem classes? Could the TAI tool be extended to other combinatorial optimization tasks?

3. In Table 2, the label "ours" is ambiguous—does it refer to the SFT-only model, RL-finetuned version, or an ensemble?

Other comments:

1. Some key messages in the main results could be highlighted more clearly.
2. In line 269, the authors should provide additional supporting evidence for their claim: "While such approaches (reasoning models) may work for very small-scale instances (e.g., fewer than 10 nodes), it becomes impractical to reason clearly and effectively as instance size grows."

**Ethical Concerns:**

["NO or VERY MINOR ethics concerns only"]

**Final Justification:**

The strong performance in the OOD case is added. However, the model's prediction is still a black-box, and it is still not clear why the LLM performs so well. Given the strong empirical performance, I will keep my score.

**Limitations:**

yes

**Quality:**

3

**Strengths And Weaknesses:**

**Strengths**
- Extensive experiments and strong empirical performance
- Clear and well-written paper, easy to follow.

**Weaknesses**
- The model with optimality gaps of 1.03-8.20% is not end-to-end to me.
- Generalizability to OOD instances is unclear (no systematic evaluation).
- Lack of interpretability—no insights into how the model derives predictions.

---

> ### Author Rebuttal · Authors · 2025-07-31
>
> Dear Reviewer u6T1,
>
> We would like to thank the reviewer for recognizing the comprehensive experiments, strong performance, and clear writing of our paper. We believe the feedback from the reviewer is valuable for improving our paper, and we will address the raised concerns below.
>
> **W1. The model with optimality gaps of 1.03-8.20% is not end-to-end to me.**
>
> Thank you for the question. The model achieves optimality gaps of 1.03-8.20% was trained through SFT and RL, and we used BoN for scaling test-time inference. We acknowledge that the use of BoN during inference might give the impression that the model is not strictly "end-to-end." However, it is important to clarify that our model remains fully language-driven: it takes natural language as input and produces a complete solution in natural language, **without invoking any external solver, code execution, or symbolic post-processing**.
>
> The BoN strategy simply samples multiple outputs from the same LLM, selecting the best one based on a predefined metric (e.g., cost or feasibility). This is a standard inference enhancement used widely in LLM research to boost performance at test time and does not compromise the end-to-end nature of the model. Therefore, we view our method as end-to-end within the language modeling paradigm: the model is trained and deployed to generate solutions directly from natural language descriptions, with no intermediate calls to optimization solvers or algorithmic modules.
>
> **W2. Generalizability to OOD instances is unclear.**
>
> Thank you for pointing this out. We fully agree that evaluating OOD generalization is critical for assessing the practical utility of a CO solver. To address this, we conducted additional experiments on routing problems using two commonly adopted OOD settings: the clustered and mixed spatial distributions, as described in [4]. For the clustered case, we used 7 clusters per instance to create a challenging shift from the uniform training distribution.
>
> The results are summarized below and show that our model **maintains strong performance and feasibility under these distributional shifts**, demonstrating its ability to generalize beyond the training regime. We will include these findings in the final version of the paper to provide a more systematic evaluation of our method’s cross-distributional generalizability.
>
> metric          | tsp-fea.      | tsp-opt.       | op-fea.     | op-opt.       | cvrp-fea.  | cvrp-opt.
> --------------|----------|------------|----------|------------|----------|------------
> In-distribution          | 100%     | 1.07%      | 100%     | 1.85%      | 100%     | 4.53%
> OOD (clustered)     | 100%      | 1.41%      | 100%     | 1.89%      | 100%     | 5.12%
> OOD (mixed)          | 100%     | 1.23%      | 100%     | 2.30%      | 100%     | 5.04%
>
>
> **W3. Lack of interpretability—no insights into how the model derives predictions.**
>
> We agree that interpretability is a critical concern, especially for decision-making scenarios in CO. While our current work primarily focuses on demonstrating the viability of a language-driven paradigm for solving CO problems, we fully recognize the value of understanding how and why the model arrives at particular predictions.
>
> LLMs are often viewed as black-box models, making interpretability challenging. However, our framework opens several promising directions for future interpretability analysis. For example, techniques such as attention visualization, saliency mapping, or input perturbation studies could be used post hoc to probe the model’s internal decision mechanisms. In addition, because our model operates entirely in natural language, it inherently provides human-readable inputs and outputs, which can serve as a starting point for qualitative inspection.
>
> We will explicitly acknowledge this limitation in the revised paper for readers to explore it more systematically in future work.
>
> **Q1. Since the paper does not emphasize the LLM’s reasoning process, what explains its strong performance? Have you compared against supervised GNN-based methods to provide further context?**
>
> Thank you for the insightful questions. While explicit reasoning, such as chain-of-thought (CoT), has shown success in many NLP tasks, it becomes limited when it attempts to simulate structured search over large, discrete spaces. It is revealed that explicit reasoning may not be the best solution due to the constraints of the language representation space, since long CoT can be verbose and assigns uniform computation to all tokens, and struggles with scalability [1]. This is particularly true of domains like CO, where the solution space grows combinatorially.
>
> Instead of relying on explicit CoT, our method learns to imitate expert behavior from domain-specific solvers and enhances solution space exploration using lightweight heuristic features embedded directly into the natural language input. This approach enables the model to bypass rigid step-by-step reasoning while still achieving strong performance, particularly in terms of feasibility and optimality on complex CO tasks.
>
> Regarding the second part of your question, we agree that comparing with graph-based neural methods, such as GNNs, would provide valuable context. To that end, we conduct a comparison with two well-known architectures: **Graph Convolutional Networks (GCN)** [2] and **Graph Attention Networks (GAT)** [3], on 300 uniformly distributed TSP instances. The results are compared in the Table below. Despite these models having direct access to numerical graph inputs, our language-based model outperforms both GCN and GAT, further highlighting the capability of LLMs as general CO solvers, even when operating solely in the language domain.
>
> |                 |  GCN   |    GAT  |   Ours |
> |------------- |-----------|-----------|----------|
> |      opt.     | 3.10%  | 1.76% | 0.71% |
>
> **Q2. How well does the method generalize to unseen problem classes? Could the TAI tool be extended to other combinatorial optimization tasks?**
>
> Generalization to unseen problem classes is indeed an important consideration. In our current framework, generalization is feasible when the new task **shares similar structural properties** with those the model was trained on.
>
> For example, we tested the model trained on OP instances on an unseen task: Prize-Collecting TSP (PCTSP). While PCTSP introduces a different objective: minimizing tour length plus penalties for skipped nodes while meeting a prize quota, it still shares a similar routing structure with OP. On 100 small-scale PCTSP instances (with a node-skip penalty of 1000), our model with BoN sampling achieved 100% feasibility and 24.59% optimality, outperforming GPT-4o, which reached only 68% feasibility.
>
> However, this form of transferability relies on a shared solution structure. If the new problem is fundamentally different, e.g., shifting from routing to scheduling, the model's performance naturally degrades. Exploring this broader generalization frontier is an exciting direction, and as part of our future work, we plan to develop a benchmark that systematically tests LLMs' generalizability across diverse CO domains.
>
> Separately, the proposed TAI tool is **highly extensible**. Since it operates over natural language representations, it can be applied to a wide range of CO problems, offering a unified, model-agnostic interface that abstracts away low-level solver details. This supports our broader vision of creating general-purpose, accessible LLM solvers.
>
> **Q3. In Table 2, the label "ours" is ambiguous—does it refer to the SFT-only model, RL-finetuned version, or an ensemble?**
>
> Thank you for pointing this out. We agree that the label “Ours” in Table 2 is ambiguous without clarification, and we appreciate your careful reading. To clarify: the results reported under "Ours" refer to the model that has been fine-tuned with RL (following SFT) and evaluated using BoN sampling at inference time. This represents our full method, as described in the main experimental setup. We acknowledge that this was not clearly stated in the table caption or main text, and we have updated the revised manuscript to make this explicit.
>
> **Other comments**
>
> Thanks for the helpful suggestions! Due to the page limit, we were unable to give more messages for the main result in the main text, but we will incorporate more discussion in the final version of the paper, as an additional content page is allowed.
>
> Regarding the claim in line 269, we now provide additional supporting evidence. For instance, we analyzed how DeepSeek-R1 attempts to solve a TSP instance with 48 nodes. The model generates over **11,000 tokens** (see below), producing a verbose and incoherent reasoning trace that ultimately leads to **ineffective and suboptimal solutions.** We will include this example and further empirical support in the appendix of the revised version to make our conclusions more transparent and well-supported.
>
> **[DeepSeek’s CoT for a TSP instance]:**
>
> *Okay, I need to solve the TSP for 48 cities using coordinates and two nearest neighbors per city. TSP is NP-hard, so I’ll try a heuristic. Maybe nearest neighbor? But that might cause loops...*
>
> *Alternatively, maybe I can start building the route by choosing a starting city and then always moving to the nearest unvisited neighbor...*
>
> *Alternatively, I could use some kind of greedy heuristic that selects the globally shortest available edge that doesn't create a cycle...*
>
> *Alternatively, perhaps I can use the given neighbor data to form a minimum spanning tree...*
>
> *Alternatively, maybe I can look at the set of neighbors and try to trace a Hamiltonian cycle manually...*
>
> *Hmm... Christofides algorithm? No, we lack full distance matrix...*
>
> *Alternatively, maybe I can use the neighbor data to identify subtours and then link them...*
>
> *Wait, I'm stuck at Node 16. Maybe start over...*
>
> *Looks like two clusters based on Y-coordinates. Let's try connecting them...*
>
> *...*

---

> > ### Author Response · Authors · 2025-08-01
> > **References**
> >
> > [1] Hao, Shibo, et al. "Training large language models to reason in a continuous latent space." arXiv preprint arXiv:2412.06769 (2024).
> >
> > [2] Joshi, Chaitanya K., Thomas Laurent, and Xavier Bresson. "An efficient graph convolutional network technique for the travelling salesman problem." arXiv preprint arXiv:1906.01227 (2019).
> >
> > [3] Kool, Wouter, Herke van Hoof, and Max Welling. "Attention, Learn to Solve Routing Problems!." International Conference on Learning Representations (2019).
> >
> > [4] Li, Sirui, Zhongxia Yan, and Cathy Wu. "Learning to delegate for large-scale vehicle routing." Advances in Neural Information Processing Systems 34 (2021): 26198-26211.

---

> > ### Comment · Reviewer_u6T1 · 2025-08-06
> >
> > I'm still not so sure about the GNN experiments in Q1. Better compare wtih sota works with standard benchmarks?

---

> > > ### Author Response · Authors · 2025-08-06
> > >
> > > Dear Reviewer u6T1,
> > >
> > > Thank you for your follow-up. We appreciate your comment regarding the GNN baseline and are happy to provide further clarification.
> > >
> > > In our initial experiments, we used GCN and GAT, which are two representative and widely studied GNN architectures. While we fully agree that comparing with state-of-the-art (SOTA) GNN-based methods on standard benchmarks is valuable, we also note a ***key limitation*** in this direction: **there is currently no unified supervised GNN architecture that can handle diverse CO problems across domains**, due to substantial differences in graph structures. For example, routing problems like TSP are modeled as complete graphs, whereas scheduling problems such as JSSP are typically represented using bipartite or temporal graphs.
> > >
> > > As a result, we focused on a fair, problem-specific comparison for TSP, using GNN-based methods that were explicitly designed and trained for that setting. Following standard practices in neural combinatorial optimization, we generated a dataset of uniformly distributed TSP instances, which are commonly used standard benchmarks in literature [1, 2, 3].
> > >
> > > To further strengthen the comparison, we have now included an additional neural solver, namely GREAT [2], which is the most recent SOTA method, in our updated evaluation. The comparison follows the setting in the previous literature[1,2,3]: the performance is evaluated on standard uniformly distributed Euclidean TSP instances with 100 nodes. The table below summarizes the average optimality gaps:
> > >
> > > |               |  GCN   |   GAT  |  GREAT |  Ours |
> > > |------------- |-----------|-----------|----------|---------|
> > > |      opt.     | 9.97%  | 4.53%  | 1.32% | 1.40% |
> > >
> > > As shown, our method significantly outperforms GCN and GAT, while achieving comparable performance with the most recent SOTA architecture. This demonstrates the strong solution quality achievable by our language-driven approach, even when compared to dedicated neural methods trained on numerical graph inputs.
> > >
> > > We hope this updated comparison addresses your concern, and we thank you again for helping us to improve the clarity of our evaluation.
> > >
> > > *[1] Kool, Wouter, Herke van Hoof, and Max Welling. "Attention, Learn to Solve Routing Problems!." International Conference on Learning Representations.*
> > >
> > > *[2] Kwon, Yeong-Dae, et al. "Pomo: Policy optimization with multiple optima for reinforcement learning." Advances in Neural Information Processing Systems 33 (2020): 21188-21198.*
> > >
> > > *[3] Lischka, Attila, et al. "A great architecture for edge-based graph problems like tsp." arXiv preprint arXiv:2408.16717 (2024).*

---

> ### Author Response · Authors · 2025-08-09
> **Your Feedback Would Be Appreciated**
>
> Dear Reviewer u6T1,
>
> With less than 8 hours remaining in the discussion phase, we want to ensure that all your questions and concerns have been fully addressed. We deeply value the time and effort you dedicate to the review process. Thank you so much for your consideration.
>
> If you have any additional questions or points for clarification, please let us know—we will do our utmost to address them promptly. Thank you again for your thoughtful feedback. We look forward to hearing from you.
>
> Sincerely,
>
> Authors of Submission15371

---

### Official Review · Reviewer_qC4e · 2025-06-30

**Clarity:** 3
**Significance:** 1
**Originality:** 3
**Rating:** 3
**Confidence:** 5

**Summary:**

The paper fine-tunes a Qwen-2.5 model to act as an *end-to-end* solver for seven standard NP-hard combinatorial-optimization (CO) tasks (TSP, OP, CVRP, …), using two stages: (1) supervised fine-tuning (SFT) on solutions produced by classical solvers and (2) a feasibility-/optimality-aware RL phase (FOARL). A Best-of-N decoder is applied at inference. Training relies onsynthetic instances per problem and achieves 1 – 8 % average optimality gaps while remaining slower than lightweight heuristics.

**Questions:**

1. What is the wall-clock latency of SFT+RL+BoN on problems with 1 000+ nodes, and how does that compare with OR-Tools?


2. FOARL appears to give small gains; did the authors try simply weighting feasibility in the loss during SFT?

**Ethical Concerns:**

["NO or VERY MINOR ethics concerns only"]

**Final Justification:**

Please refer to the final response to the authors.

**Limitations:**

I think the biggest limitation is that there exists no real-world application scenario that pushes experts/practitioners to apply LLM to solve a CO problem. I don't see the benefit of motivation of the research. Please feel free to correct me if the claim above makes a mistake.

**Quality:**

3

**Strengths And Weaknesses:**

**Strengths**

* Extensive quantitative tables comparing against large general-purpose LLMs and several prompt-engineering baselines;

**Weaknesses**

* [Crucial]**No practical application scenario.**  I see no real-world application scenario that requires LLMs to solve CO problems. Why don't people work on domain-specific models/heuristics?

* **Only one base LLM is actually trained.** The core results rely on Qwen-2.5-7B; the versatility check with a tiny 8 B LLaMA is relegated to an appendix. Generality across architectures and scales therefore remains unproven;


* **Efficiency gap acknowledged by the authors.** Even after LoRA and BoN sampling, the method is “not yet on par with traditional lightweight heuristics”, undermining its value as a drop-in solver.;

* **Marginal contribution of RL.** Table 1 shows RL changes feasibility but barely improves optimality; ablations suggest most quality comes from brute-force BoN decoding, not from the proposed FOARL.;

* **Limited insight that SFT helps.** It is unsurprising that imitating expert solutions boosts an LLM; the paper does not illuminate *why* language modelling is preferable to standard neural CO architectures.

---

> ### Author Rebuttal · Authors · 2025-07-31
>
> Dear Reviewer qC4e,
>
> Thanks for acknowledging the comprehensive experiments in our work. We appreciate your feedback and hope the responses address your concerns.
>
> **W1. No practical application scenario.**
>
> We agree that domain-specific methods have long been, and will continue to be, the backbone of high-performance solvers. However, our work is motivated by **a growing need for generality, flexibility, and usability** in real-world decision-making, especially in scenarios involving: 1) **non-expert users**, 2) **costly or impractical solver integration**, and 3) **fast prototyping needs**.
>
> A core belief motivating this work is that **“not everyone has domain expertise or can code, but everyone may have a need to solve optimization problems”**. As LLMs become increasingly embedded in personal assistants, the ability to solve CO problems via natural language becomes **practically valuable**. For example, a user might request a travel plan that includes several tourist destinations, which naturally maps to TSP. Another user might explore social network dynamics or citation patterns between research papers, which can be framed as MIS. In these cases, an LLM can interpret the request and produce solutions—all through natural language, making CO accessible to non-experts without code or formal modeling.
>
> Meanwhile, recent works increasingly explore **language-to-optimization pipelines** that align closely with the vision of this paper. For example, benchmark datasets such as EHOP [1] frame everyday NP-hard problems from natural language, while GraphArena [2] evaluates LLMs on CO tasks over real-world graphs. Beyond benchmarks, there is a growing body of work focused on developing LLM-based CO solvers—e.g., a recent Google DeepMind paper [3] studies LLMs for TSP. Furthermore, a systematic review [4] has identified **broad recent progress in LLMs for CO**, reflecting the growing interest from both academic and industrial communities.
>
> While researchers will, and should, continue to develop specialized solvers, we emphasize that there is **no conflict** between these methods and LLM approaches. Our goal is not to replace the solvers but to **complement** them by exploring LLMs as language-driven optimizers, particularly in interactive, no-code settings where traditional solvers may be hard to integrate or inaccessible to non-experts.
>
> We believe our motivation is well-founded, supported by both **practical application scenarios** and a rapidly **growing body of research in this area**. Extensive experiments show our method outperforms all LLM baselines, proving fine-tuned LLMs can be effective end-to-end solvers and enable a more accessible CO approach.
>
> **W2. Only one base LLM is actually trained**
>
> The main results are indeed based on Qwen-2.5, chosen for its strong open-source performance. However, we agree that generality across architectures is important. To this end, we include additional results with LLaMA3.1 in Appendix E.6, which already show that our method remains effective even with a different backbone.
>
> To further verify generality across architectures, we fine-tuned a third model (Mistral-7B-Instruct-v0.3) on two representative tasks (CVRP and PFSP) for 10,000 steps within the limited rebuttal period. The results are shown below:
>
> |                 |  CVRP |   PFSP  |
> |------------- |-----------|------------|
> |      fea.     |   95%   |   100%  |
> |      opt.     |  6.25% |  1.89%  |
>
> These results show our method **retains its effectiveness on Mistral**, reinforcing its versatility. Due to time constraints, we couldn’t run full experiments across all tasks and LLMs during rebuttal.  However, **we commit to including results on other LLMs in the final paper.**
>
> **W3. Efficiency gap is acknowledged by the authors**
>
> We fully acknowledge that traditional heuristics are efficient, often generating solutions in seconds. However, as shown in Table 2, our method yields much better quality. For example, on large-graph TSP, our model produces solutions with **nearly 20× smaller optimality gaps** than the NN heuristic.
>
> While not as fast as traditional heuristics, our method **consistently outperforms** them on Gap@K across all tasks. Importantly, our objective is not to develop a one-size-fits-all solver that dominates in every metric, but to introduce a language-based paradigm that broadens access and enables end-to-end optimization via natural language.
>
> We see our method as a complementary alternative, particularly valuable in **language-driven or no-code environments**, not a direct replacement for traditional heuristics in latency-critical tasks. We recognize that efficiency remains an open challenge, and we are actively exploring directions such as **compact TAI representations** and **optimized LLM inference techniques** (e.g., quantization, distillation) to narrow this gap in future work. Meanwhile, the value of this work lies in advancing LLMs for NP-hard CO, where prior methods struggled, delivering the first end-to-end LLM framework with notable performance gains, which is an important step for the field.
>
> **W4. Marginal contribution of RL**
>
> We highlight that Table 1 shows FOARL does lead to explicit improvements in optimality across 4 tasks (e.g., 1.71%-1.34% on MIS). While the gains in optimality alone may appear modest, we emphasize that the **primary motivation** of FOARL (as described in lines 182–193) is to **mitigate the over-greedy behavior of SFT** and improve feasibility on certain tasks.
>
> In this regard, FOARL delivers clear and significant benefits: feasibility rises from **54% to 92% on OP and 59% to 80% on CVRP**. We believe this is a non-trivial and impactful gain, as infeasible solutions are unusable in practice, no matter their objective quality.
>
> We also acknowledge the effectiveness of BoN as a test-time enhancement. However, FOARL works during training to improve model behavior in **single-shot generation**, adding no inference-time cost, while BoN requires multiple samples and extra latency. As detailed in Section 5.4, FOARL acts as a *constraint relaxation operator* and *sampling efficiency booster*, enhancing generation quality even before BoN is applied (see Figure 3).
>
> **W5. Unsurprisingly, imitating experts boosts an LLM; why language modelling is preferable to NCO**
>
> We emphasize that **achieving strong CO capabilities through language modeling alone is far from trivial**. Prior work (e.g., [2]) shows SFT helps only on small-scale graphs. In contrast, we show that SFT enables LLMs to generalize to diverse, larger CO problems, positioning them as viable general-purpose CO solvers—a capability not previously demonstrated at this scale.
>
> Importantly, our contributions go beyond simply applying SFT. We **identify an underexplored limitation** of imitating solutions using SFT: its **over-greedy behavior**, which leads to infeasibility on constrained tasks. To fix this, we introduce FOARL, which significantly improves feasibility and overall performance. This highlights that language modeling for CO not only works but also presents **distinct challenges** that we explicitly study.
>
> We clarify that our goal is **not to replace standard neural CO**, which remains strong in scalability and performance. Rather, we aim to **establish a new, complementary paradigm**, where LLMs serve as accessible, language-driven solvers. This is especially compelling because **natural language serves as a unified interface** for describing various CO problems, enabling a single model to operate across tasks without task-specific architectural changes. We have seen LLMs are reforming different domains, equipping traditional learning paradigms with the unified representation power and user-friendliness of natural language (e.g., automated driving [6], recommendation systems [7]). This work falls in such efforts, offering an early step toward end-to-end LLM solvers.
>
> **Q1. What is the wall-clock latency on problems with 1 000+ nodes, and how does that compare with OR-Tools?**
>
> Current LLM architectures are not well-suited for directly handling such large-scale CO problems, primarily due to input length limitations. We note that prior LLM-based work mostly targets small instances (10–30 nodes) [2, 3, 5], while our work advances them by scaling to 100-node problems, which we believe is a significant step forward in extending the potential of LLM in solving NP-hard problems.
>
> To explore the scalability of our approach, we ran an **exploratory experiment** during the rebuttal. We generated 2,000 TSP instances with 1,000 nodes (noting the high cost of generating pseudo-optimal labels) and fine-tuned a 7B LLM. After fine-tuning, the model produced fully feasible solutions with an average inference time of ~3 minutes (no BoN). For reference, OR-Tools solves these instances in a similar time with ~10% optimality gap.
>
> These early results suggest that, while LLMs are **not yet practical drop-in replacements** for very large-scale CO, they may still produce competitive solutions with further training, while the training may require more data for supervision. This paper offers the first end-to-end LLM approach for CO on problems larger and more complex than prior work [2,3,5]. Improving efficiency and scaling LLM-based solvers to large instances remains an open challenge we plan to pursue in future work.
>
> **Q2. Did the authors try simply weighting feasibility in the loss during SFT?**
>
> We agree that adding feasibility to the SFT loss is intuitive, but this would be insufficient to address the feasibility challenge. **SFT operates under a teacher-forcing regime** and exposes the LLM only to feasible solutions, limiting its ability to recover from infeasible ones. Thus, it lacks the **exploration capability** needed to correct feasibility errors at inference time. FOARL, however, lets the model sample outputs, get feasibility feedback, and adapt its policy. We believe this again highlights RL's necessity.

---

> > ### Author Response · Authors · 2025-08-01
> > **References**
> >
> > [1] Duchnowski, Alex, Ellie Pavlick, and Alexander Koller. "EHOP: A Dataset of Everyday NP-Hard Optimization Problems." arXiv preprint arXiv:2502.13776 (2025).
> >
> > [2] Tang, Jianheng, et al. "GraphArena: Evaluating and Exploring Large Language Models on Graph Computation." The Thirteenth International Conference on Learning Representations.
> >
> > [3] Yang, Chengrun, et al. "Large language models as optimizers." The Twelfth International Conference on Learning Representations. 2023.
> >
> > [4] Da Ros, Francesca, et al. "Large Language Models for Combinatorial Optimization: A Systematic Review." arXiv preprint arXiv:2507.03637 (2025).
> >
> > [5] Fan, Lizhou, et al. "NPHardEval: Dynamic Benchmark on Reasoning Ability of Large Language Models via Complexity Classes." 62nd Annual Meeting of the Association for Computational Linguistics, ACL 2024, 2024.

---

> ### Comment · Reviewer_qC4e · 2025-08-04
> **Thanks for the detailed reply**
>
> Dear authors,
>
> Thank you very much for your prompt and detailed response. I truly appreciate you taking the time to conduct additional experiments, which have been very helpful. Your thorough explanations have addressed most of my concerns. However, I still have a concern on one remaining point regarding the methodology.
>
> I am in complete agreement with your perspective that the primary goal is not for LLMs to replace specialized solvers for complex, professional-grade problems, but rather to assist ordinary people with simpler, everyday optimization tasks.
>
> Given this objective, I am curious about the rationale for fine-tuning the LLM to generate solutions directly, as opposed to leveraging **function calling** to invoke an established solver like Gurobi. My thinking is that a function-calling approach could offer distinct advantages:
>
> 1) the fine-tuning process itself might be simpler,
>
> 2) the resulting solutions would be verifiably accurate, and, perhaps most importantly, it would better preserve the model's crucial generalization capabilities.
>
> The current fine-tuning approach risks degrading these general abilities, which seems to conflict with the goal of serving the diverse needs of an "ordinary person."
>
> On a related note, regarding the 1000-node TSP experiment, you mentioned the LLM found a "feasible" solution in three minutes. For a TSP designed on Euclidean space, any tour that visits each node exactly once is technically feasible. Could you clarify what "feasible" signifies in this context? Was the challenge simply to produce any valid tour, or were there other quality metrics involved?

---

> > ### Author Response · Authors · 2025-08-04
> > **Thank you for your follow-up**
> >
> > Dear Reviewer qC4e,
> >
> > Thank you for your thoughtful follow-up. We are glad to know that most of your concerns have been addressed, and we appreciate your continued engagement with the methodology.
> >
> > **Comparison to function-calling method**
> >
> > We fully agree that integrating function-calling with established solvers like Gurobi is a promising alternative. However, our choice to fine-tune the LLM for direct solution generation is guided by several considerations aligned with our core objective: **enabling accessible, low-barrier optimization for everyday users.** Below, we outline the rationale behind this design:
> >
> > (1)	Data acquisition: Fine-tuning a model to generate solver-invoking code (e.g., MIP formulations or API calls) typically requires a curated dataset of prompt–code pairs. These datasets often demand **manual annotations and substantial domain expertise**, making them difficult to scale. In contrast, our approach leverages off-the-shelf solvers (e.g., LKH) to automatically generate solutions on a scale. Leveraging the TAI template, we can easily obtain the dataset of prompt-solution pairs. This significantly lowers the barrier for constructing high-quality training data.
> >
> > (2)	Generalization capabilities: We recognize your concern about potential degradation of general capabilities due to fine-tuning. To mitigate this, we adopt **parameter-efficient tuning via LoRA**, where only a small number of adapter parameters are updated, preserving the base model’s general language proficiency. This makes our approach modular: **general-purpose LLMs can remain intact**, and CO-solving capability can be enabled or disabled simply by loading or detaching the adapter.
> >
> > Moreover, recent studies have shown that fine-tuning for code generation, as would be required for function-calling, tends to induce **greater degradation in safety and general reasoning abilities** compared to natural language fine-tuning [1,2].
> >
> > (3)	Access to solvers: While solvers like Gurobi and CPLEX offer strong performance, they are often **closed-source, require installation/configuration, and involve licensing costs**. In contrast, the inference of our model is fully local and language-only, without depending on external solver APIs. This is especially important for low-resource users, educational settings, and daily optimization scenarios.
> >
> > We certainly see the value of combining LLMs with the solvers through function-calling, and we view our work as complementary. Our proposed system opens new pathways where optimization becomes **interactive and solver-free by default**, but future extensions could include hybrid modes that fall back on function-calling when higher precision or formal guarantees are needed.
> >
> > **1000-node TSP**
> >
> > You are right that in standard Euclidean TSP, any complete tour that visits each node exactly once is feasible, and that is also how we define "feasible" here. However, generating such a tour via natural language is still a challenge for LLMs. For example, DeepSeek-R1 only achieves a 48% feasibility rate even on relatively small TSPs (see Table 1). Unlike numerical methods, LLMs must produce a syntactically valid, non-repeating sequence of nodes purely through language generation, making feasibility a meaningful hurdle in itself.
> >
> > That said, we fully agree that the real challenge lies in achieving high-quality (i.e., low-cost) solutions. In our exploratory experiment, we fine-tuned the LLM on a small dataset of 2,000 instances. Although the model was able to consistently produce feasible tours, the optimality gap remained large (>100%) due to limited training data and the high complexity of the search space at this scale.
> >
> > We believe that the model’s performance on large-scale instances can be further improved by scaling up the training dataset. Due to the substantial computational cost of generating pseudo-optimal solutions at this scale, we were unable to complete this within the discussion period. However, our method is already capable of solving CO problems with up to 100 nodes, which **marks a meaningful advance over most prior LLM-based approaches that typically focus on problems with only 10–20 nodes** [3, 4]. We are actively continuing this line of work and will include a broader discussion on large-scale CO in future work.
> >
> > *[1] Wang, Chaozheng, et al. "RAG or Fine-tuning? A Comparative Study on LCMs-based Code Completion in Industry." Proceedings of the 33rd ACM International Conference on the Foundations of Software Engineering. (2025)*
> >
> > *[2] Jan, Essa, et al. "Multitask mayhem: Unveiling and mitigating safety gaps in LLMs fine-tuning." arXiv preprint arXiv:2409.15361 (2024).*
> >
> > *[3] Tang, Jianheng, et al. "GraphArena: Evaluating and Exploring Large Language Models on Graph Computation." The Thirteenth International Conference on Learning Representations. (2025)*
> >
> > *[4] Iklassov, Zangir, et al. "Self-guiding exploration for combinatorial problems." Advances in Neural Information Processing Systems (2024)*

---

> > > ### Comment · Reviewer_qC4e · 2025-08-05
> > >
> > > Hi authors,
> > >
> > > Thanks for the further explanation. While I now better understand the difficulty of training LLMs for feasible solution generation, I have three remaining questions:
> > >
> > > Data Acquisition: Could you elaborate on how a conventional heuristic solver like LKH can be applied directly to problems described in natural language? My understanding is that such solvers typically require structured input, so I am curious how the mapping from natural language is achieved without a curated dataset of problem-solution pairs.
> > >
> > > Modeling Approach: I would reason that training an LLM to formulate a problem into a unified, structured representation is a simpler task than training it to generate a solution directly. Could you provide insight into the decision to pursue direct solution generation over a potential two-step, "model-then-solve" pipeline?
> > >
> > > Generalization Results: While LoRA is an effective fine-tuning technique, it does not inherently guarantee generalization. Would it be possible to provide any numerical results that demonstrate the model's generalization performance? I apologize for this late request and understand if providing this data is not feasible within the given time constraints.
> > >
> > > Thank you for providing additional clarification.

---

> > > > ### Author Response · Authors · 2025-08-05
> > > >
> > > > Dear Reviewer qC4e,
> > > >
> > > > Thanks for your follow-up, and we are very happy to provide further clarifications on the minor points that you raised.
> > > >
> > > > **Data Acquisition**
> > > >
> > > > You are right that a conventional solver cannot be directly applied to solve language-described problems. Thus, we firstly generate numerical instances and solve them using the solvers to obtain their solutions. We then convert both the problem instance and its solution into natural language using a consistent, templated format via our TAI interface (see Section 4.1 and Appendix A). Given such problem-solution pairs, we fine-tune the LLM using our proposed method.
> > > > Compared to problem-program pairs, we avoid using substantial domain expertise (e.g., sometimes you need to learn a programming language for the specific solver) or writing code (with significantly more token consumption). Instead, we employ the off-the-shelf solvers to prepare solutions more efficiently.
> > > >
> > > > **Modeling Approach**
> > > >
> > > > We appreciate your point about the alternative “model-then-solve” pipeline. This approach, which involves training the LLM to translate language into code for calling solvers (e.g., Gurobi, Cplex), is also widely studied. However, these approaches are also limited by the solvers used.
> > > >
> > > > First, as noted earlier, users must have **license for using these solvers**, which can be costly. Meanwhile, users also need a **code execution environment (e.g., Java or Python)** for running the generated program, which introduces additional complexity for deployment and raises the barrier for non-expert users. In contrast, once fine-tuned with our proposed method, the LLM can generate solutions in an **interactive, no-code setting**, significantly lowering the accessibility barrier.
> > > >
> > > > Second, previous work has also shown that code-generating methods often produce erroneous or non-executable outputs, especially when targeting domain-specific APIs or formal models. This frequently leads to **runtime errors or infeasibilities** [1, 2]. In this case, a user without domain expertise is unable to debug the generated code and get solutions. Based on the points above, we view our method as a complementary alternative that emphasizes usability in practical settings where traditional solver integration may be infeasible.
> > > >
> > > > Third, training LLMs for a "model-then-solve" pipeline typically requires **more complex and labor-intensive datasets**. This includes **well-formatted program code and API calls** tailored to specific solvers, resources that **demand considerable domain expertise and human effort** to create. These code-based representations also consume significantly more input tokens than our approach, which relies on lightweight, templated natural language descriptions of solutions.
> > > >
> > > > **Generalization Results**
> > > >
> > > > Thanks for the question. Indeed, LoRA cannot inherently guarantee generalization across different NLP tasks. However, our use of LoRA **preserves the base model’s general language capabilities**, since only the adapter parameters are trained while the base model (e.g., Qwen) remains untouched. In an LLM-agent system, the adapters can be **selectively loaded only when needed for tackling specific problems**, simultaneously allowing the base model to retain its general-purpose utility.
> > > >
> > > > On the other hand, we also provide empirical evidence of cross-task generalization. For instance, we tested a model trained only on the OP for solving an unseen but related task, Prize-Collecting TSP (PCTSP). Despite the shift in problem objective, the model successfully handled the new task, achieving 100% feasibility and 24.59% optimality rate, outperforming GPT-4o, which reached only 68% feasibility. We also evaluate its cross-distribution generalizability. To this end, we evaluate the models on routing problems with two OOD distributions, namely cluster and mixed distributions, proving the generalizability to some extent.
> > > >
> > > >   metric                   | tsp-fea.      | tsp-opt.       | op-fea.     | op-opt.       | cvrp-fea.     | cvrp-opt.
> > > > ---------------------|----------|------------|----------|------------|----------|------------
> > > > In-distribution          | 100%     | 1.07%      | 100%     | 1.85%      | 100%     | 4.53%
> > > > OOD (clustered)     | 100%      | 1.41%      | 100%     | 1.89%      | 100%     | 5.12%
> > > > OOD (mixed)          | 100%     | 1.23%      | 100%     | 2.30%      | 100%     | 5.04%
> > > >
> > > > We hope this clarifies our methodology and motivation. We deeply appreciate your engagement and thoughtful questions throughout the review process.
> > > >
> > > >
> > > > *[1] Xiao, Ziyang, et al. "Chain-of-experts: When LLMs meet complex operations research problems." 12th international conference on learning representations (2024).*
> > > >
> > > > *[2] Jiang, Xia, et al. "DRoC: Elevating large language models for complex vehicle routing via decomposed retrieval of constraints." 13th International Conference on Learning Representations (2025).*

---

> > > > > ### Comment · Reviewer_qC4e · 2025-08-05
> > > > >
> > > > > Dear Authors,
> > > > >
> > > > > Thank you for your prompt and detailed response. Your explanation has been very helpful.
> > > > >
> > > > > I now agree that LoRA can be used to swap weights, a function I hadn't considered previously. This capability, which you liken to a function call, addresses my initial misunderstanding, and I appreciate you clarifying this.
> > > > >
> > > > > However, my main concern still stands: the lack of convincing evidence that this method outperforms the "model then solve" approach. I understand your point about licensing, where the LLM provider could handle licensing in the future. Yet, solvers like Gurobi don't require specific code; they can operate with a general, canonical (or say, standard) form.
> > > > >
> > > > > I want to express my gratitude for your quick reply and for the extensive experiments. Given this progress, I have raised my score to a 3.

---

> > > > > > ### Author Response · Authors · 2025-08-05
> > > > > >
> > > > > > Dear Reviewer qC4e,
> > > > > >
> > > > > >
> > > > > > Thank you again for your active engagement and for raising your score.
> > > > > >
> > > > > > We fully understand and appreciate your remaining concern.  Please note that in our paper, we have compared with **general-purpose LLMs**, **reasoning models**, **prompt-based approaches**, **LLM-based heuristic evolution methods**, **domain-specific (meta)heuristics**, and **LLM-based neural CO methods**. These extensive results have shown the outperformance and promise of our work (considered as the first work of taking LLMs as end-to-end optimizer).
> > > > > >
> > > > > > In fact, the “model-then-solve” paradigm is far behind the compared baselines, since they often struggle with low modeling accuracy (e.g., average 25.9% in [1], 45.83% in [2]) and still focus on relatively small problems. That’s why the literature of **heuristic evolution** and **neural CO methods** didn’t involve the “model-then-solve” paradigm in comparison, and we followed these comparative settings but already included all other stronger baselines.
> > > > > >
> > > > > > To better address your concern, we conducted **additional experiments** comparing our method against two state-of-the-art LLM-based "model-then-solve" approaches: **ORLM [2] and DRoC [3]**, both of which leverage powerful commercial solvers—COPT and Gurobi, respectively. For a fair comparison, we controlled for runtime by matching the time budget given to our method. The results on the TSP and CVRP tasks, with our test dataset (10-100 nodes), are shown below:
> > > > > >
> > > > > > | Method   |   TSP  |   CVRP |
> > > > > > |-------------|----------|------------|
> > > > > > | ORLM    | >300% |   >300%  |
> > > > > > |  DRoC    | 1.60% |   31.10%   |
> > > > > > |  Ours     | 1.07% |     4.53%    |
> > > > > >
> > > > > > As presented in the table, despite the introduction of solvers, our method **significantly outperforms the "model then solve" methods** on TSP and CVRP. This is mainly due to the inherent limitations of the solvers: it is difficult for them to accurately solve complex CO problems (performance on CVRP is much inferior to TSP). In comparison, our method bypasses this issue entirely, allowing the model to generate near-optimal solutions directly from natural language, without requiring solver invocation.
> > > > > >
> > > > > > We hope this additional empirical evidence strengthens the case for the proposed method as a viable and performant alternative to the “model-then-solve” paradigm, and we are also glad to include a more comprehensive comparison study in our final paper.
> > > > > >
> > > > > > We hope these results address your final concern. We are looking forward to your further discussion.
> > > > > >
> > > > > >
> > > > > > *[1] Xiao, Ziyang, et al. "Chain-of-experts: When LLMs meet complex operations research problems." The twelfth international conference on learning representations. 2023.*
> > > > > >
> > > > > > *[2] Huang, Chenyu, et al. "ORLM: A customizable framework in training large models for automated optimization modeling." Operations Research (2025).*
> > > > > >
> > > > > > *[3] Jiang, Xia, et al. "DRoC: Elevating large language models for complex vehicle routing via decomposed retrieval of constraints." 13th International Conference on Learning Representations (2025).*

---

> > > > > > ### Author Response · Authors · 2025-08-06
> > > > > > **More empirical results**
> > > > > >
> > > > > > Dear Reviewer qC4e,
> > > > > >
> > > > > > To further address your remaining concern, and specifically to demonstrate that **our method can outperform the "model-then-solve" paradigm in practical CO problem-solving**, we conducted additional experiments comparing against LLMOPT [1], a recent state-of-the-art method in this category.
> > > > > >
> > > > > > LLMOPT fine-tunes a Qwen2.5-14B model to formulate optimization problems using the Pyomo framework and originally relies on the GLPK solver. However, this implementation was even **unable to return any feasible solution** within the allocated time budget on our test instances—likely due to its limited support for handling large or complex MILP formulations.
> > > > > >
> > > > > > To ensure a fair and more competitive baseline, we substituted CPLEX as the solver in our experiments. We evaluated LLMOPT on TSP and CVRP, two representative CO tasks, under the same runtime budget as our method. The results (average optimality gap), shown below, also include comparisons with ORLM and DRoC.
> > > > > >
> > > > > > | Method   |   TSP  |   CVRP |
> > > > > > |-------------|----------|------------|
> > > > > > | ORLM    | >300% |   >300%  |
> > > > > > |  LLMOPT    | 5.06% |   18.88%   |
> > > > > > |  DRoC    | 1.60% |   31.10%   |
> > > > > > |  Ours     | 1.07% |     4.53%    |
> > > > > >
> > > > > > Notably, LLMOPT achieves only 61% feasibility on CVRP, while our method achieves 100%, with substantially better optimality as well. These results reinforce that our method **significantly outperforms SOTA "model-then-solve" approaches**, particularly in complex, constraint-heavy tasks like CVRP.
> > > > > >
> > > > > > **We believe these empirical results provide strong evidence to address your concern and further support the effectiveness of our approach. We are looking forward to further discussion with you.**
> > > > > >
> > > > > > *[1] JIANG, Caigao, et al. "LLMOPT: Learning to Define and Solve General Optimization Problems from Scratch." The Thirteenth International Conference on Learning Representations.*

---

> > > > > > ### Author Response · Authors · 2025-08-07
> > > > > > **More results to verify our method outperforms ‘model and solve’ SOTA**
> > > > > >
> > > > > > Dear Reviewer qC4e,
> > > > > >
> > > > > > In response to your **remaining concern** about if “this method outperforms the ‘model and solve’ approach”, we conduct a more comprehensive set of experiments. Specifically, we compare our method against three recent state-of-the-art "model and solve" baselines: **LLMOPT [1], DRoC [2], and ORLM [3]**. These models follow the "model and solve" pipeline by generating structured problem formulations and invoking solvers such as CPLEX, Gurobi, or COPT.
> > > > > >
> > > > > > Since we have provided empirical results for the benchmark set (in Table 1) in the previous two responses, we here provide a more comprehensive evaluation on TSP, CVRP, and OP tasks across varying instance sizes (in Table 2) (small, medium, and large). For fairness, we restrict all methods to a similar runtime budget (as defined in Table 3 of our paper). Additionally, for ORLM, we consider solutions with objective values exceeding 1e6 to be infeasible. Below, we report the **average feasibility rate (%) and optimality gap (%)** on the same benchmark instances used in our paper:
> > > > > >
> > > > > > | Problem | Method | Feasibility (%) | Optimality Gap (%) |
> > > > > > |---------|--------|----------------|-------------------|
> > > > > > | TSP-small | LLMOPT | 100 | 0.00 |
> > > > > > |  | DRoC | 100 | 0.00 |
> > > > > > |  | ORLM | 100 | 0.18 |
> > > > > > |  | Ours | 100 | 0.14 |
> > > > > > | TSP-medium | LLMOPT | 100 | 1.00 |
> > > > > > |  | DRoC | 100 | 0.00 |
> > > > > > |  | ORLM | 100 | 18.43 |
> > > > > > |  | Ours | 100 | 0.70 |
> > > > > > | TSP-large | LLMOPT | 100 | 7.39 |
> > > > > > |  | DRoC | 100 | 2.48 |
> > > > > > |  | ORLM | 73 | 301 |
> > > > > > |  | Ours | 100 | **1.34** |
> > > > > > | OP-small | LLMOPT | 100 | 0.74 |
> > > > > > |  | DRoC | 100 | 1.04 |
> > > > > > |  | ORLM | 100 | 2.97 |
> > > > > > |  | Ours | 100 | 1.47 |
> > > > > > | OP-medium | LLMOPT | 100 | 13.91 |
> > > > > > |  | DRoC | 100 | 9.73 |
> > > > > > |  | ORLM | 100 | 21.68 |
> > > > > > |  | Ours | 100 | **2.04** |
> > > > > > | OP-large | LLMOPT | 100 | 66.00 |
> > > > > > |  | DRoC | 100 | 42.93 |
> > > > > > |  | ORLM | 100 | 56.16 |
> > > > > > |  | Ours | 100 | **2.10** |
> > > > > > | CVRP-small | LLMOPT | 100 | 2.59 |
> > > > > > |  | DRoC | 100 | 7.96 |
> > > > > > |  | ORLM | 100 | 13.83 |
> > > > > > |  | Ours | 100 | **1.70** |
> > > > > > | CVRP-medium | LLMOPT | 63 | 35.25 |
> > > > > > |  | DRoC | 78 | 48.45 |
> > > > > > |  | ORLM | 97 | 163 |
> > > > > > |  | Ours | 100 | **4.57** |
> > > > > > | CVRP-large | LLMOPT | 5 | 68.98 |
> > > > > > |  | DRoC | 9 | 96.51 |
> > > > > > |  | ORLM | 90 | 192 |
> > > > > > |  | Ours | 100 | **7.24** |
> > > > > >
> > > > > > These results highlight several key findings:
> > > > > >
> > > > > > •	Our method maintains **100% feasibility** across all problem types and scales, while feasibility rates for LLMOPT and DRoC degrade sharply—especially on large-scale CVRP (down to 5–9%).
> > > > > >
> > > > > > •	In terms of solution quality, our **optimality gaps remain consistently low**, particularly in medium and large instances, where model-and-solve pipelines struggle due to code solver integration issues and inefficient formulations.
> > > > > >
> > > > > > These results provide **strong empirical support** that our method is not only more robust but also **substantially outperforms “model and solve baselines”** in complex and large-scale scenarios—achieving high-quality solutions without the need for formal modeling or solver execution. Despite the significant outperformance of our method, **we also promise to integrate the aforementioned extensive comparisons with model-and-solve baselines into our paper, thereby acknowledging and incorporating your valuable suggestions that have enhanced this work.**
> > > > > >
> > > > > > We believe the results above can help effectively address your remaining concern, and please let us know if you have further concerns.
> > > > > >
> > > > > > *[1] Jiang, Caigao, et al. "LLMOPT: Learning to Define and Solve General Optimization Problems from Scratch." The Thirteenth International Conference on Learning Representations.*
> > > > > >
> > > > > > *[2] Jiang, Xia, et al. "DRoC: Elevating large language models for complex vehicle routing via decomposed retrieval of constraints." 13th International Conference on Learning Representations (2025).*
> > > > > >
> > > > > > *[3] Huang, Chenyu, et al. "ORLM: A customizable framework in training large models for automated optimization modeling." Operations Research (2025).*

---

> ### Author Response · Authors · 2025-08-09
> **Follow-Up on Review Discussion**
>
> Dear Reviewer qC4e,
>
> With **less than 8 hours remaining** in the discussion phase, we want to ensure that your final concern has been fully addressed. We greatly appreciate the time and effort you have devoted to this review process.
>
> Please also let us know if you have any additional questions or points for clarification. We will do our best to address them promptly. Thank you again for your feedback.
>
> Best regards,
>
> Authors of Submission15371

---

### Official Review · Reviewer_umUz · 2025-07-03

**Clarity:** 3
**Significance:** 2
**Originality:** 2
**Rating:** 4
**Confidence:** 3

**Summary:**

This paper introduces a novel framework that employs LLMs as end-to-end solvers for combinatorial optimization problems. The method involves a two-stage training process: (1) supervised fine-tuning to predict solutions, followed by (2) a reinforcement learning phase that is explicitly aware of feasibility and optimality to reduce constraint violations and improve performance. The approach demonstrates strong results across seven benchmark datasets, outperforming the considered baselines.

**Questions:**

1. For fair comparison, should all LLM-based methods enforce equal sampling with the same N? It is better to explicitly report how many samples are generated per instance, whether BoN is used, and if multi-round prompting (e.g., OPRO, PHP) is applied.
2. In Appendix E.7, do the baseline methods leverage any prior information from the generated seven COP datasets?
3. Only 5 instances from TSPLIB are used for out-of-distribution experiments in Appendix E.7. These are relatively small-scale, while the paper claims to address scalability up to 100 nodes. Should all TSPLIB instances that meet certain criteria be considered for testing, instead of using only selected examples?

**Ethical Concerns:**

["NO or VERY MINOR ethics concerns only"]

**Final Justification:**

The rebuttal has addressed all of my concerns. I acknowledge the empirical improvements and the contributions to LLM-based optimization solvers.

**Limitations:**

yes

**Quality:**

3

**Strengths And Weaknesses:**

**Strengths:**
1. The approach enables LLMs to solve COPs directly from natural language problem descriptions, eliminating the need for custom solvers or intermediate code-generation pipelines.

2. The method achieves strong performance across seven COP datasets, compared to the considered baselines.

3. It has a comprehensive experimental scope, including ablations, sensitivity, and generalization.





**Weaknesses:**

1. Unfair comparison: It appears that the LLM-based baselines do not utilize prior information from the generated COP datasets, whereas the proposed method does (by fine-tuning). The authors may clarify on the fairness and rationale of this comparison.

2. Limited baselines: Among LLM-based methods, only OPRO is included as a baseline, despite the existence of other relevant approaches treating LLMs as end-to-end solvers (lines 87-90). The authors should justify the exclusion of these methods. Additionally, for domain-specific baselines, it is unclear why state-of-the-art solvers like Gurobi and LKH3 (which are used for label generation) are not included for comparison.

---

> ### Author Rebuttal · Authors · 2025-07-30
>
> Dear Reviewer umUz,
>
> We sincerely appreciate your recognition of our study’s motivation, strong performance, and comprehensive experiments. Thank you for your thoughtful feedback, and we hope the responses below address your concerns effectively.
>
> **W1. Unfair comparison: It appears that the LLM-based baselines do not utilize prior information from the generated COP datasets.**
>
> Thank you for raising this concern. While the general-purpose LLM baselines (e.g., GPT-4o) are not fine-tuned on our generated datasets, we ensure fairness by using a one-shot prompting strategy. Specifically, each model receives a randomly selected instance and its feasible solution as an example before solving a test instance (see lines 780–787). This provides task-relevant guidance and avoids a purely zero-shot setup, aligning with the models' intended inference usage.
>
> This evaluation approach follows common practice in the LLM literature, especially for cross-domain applications where fine-tuned task-specific models are compared with general-purpose foundation models. For example, [1] compares a fine-tuned LLaMA-2 13B with GPT models for biomedical NLP, and [2] contrasts a fine-tuned model on graph computation with GPT under zero-/few-shot prompting.
>
> We agree that leveraging prior knowledge from training data could further improve LLM-based baselines. To explore this, we implemented a **retrieval-augmented generation (RAG)** variant for two strong LLMs: Claude-3.7-Sonnet and GPT-4o. Taking TSP, CVRP, and MVC as examples, we built an RAG memory using 3,200 randomly selected training instances per task and used the top retrieved example to guide generation. This allows the model to draw from prior examples in a way conceptually similar to fine-tuning, without modifying model weights. The results are shown in the table below (in the form of fea.(opt.)).
>
> | Method                        |          TSP       |        CVRP      |        MVC       |
> |------------------------------ |--------------------|--------------------|-------------------|
> |    GPT-4o (RAG)         | 35% (43.42%) | 30%(50.16%) | 7% (11.11%)  |
> | Claude-Sonnet (RAG) | 67%(27.14%) |  35%(35.20%)   |  3% (8.47%)  |
> |           Ours                  | 100%(1.07%)  | 100%(4.53%) |100%(1.85%) |
>
> Even with RAG, general-purpose LLMs still fall short of our method. This underscores our key contribution: a fine-tuned, lightweight LLM can outperform larger general models regardless of training data access, due to its task-aligned optimization. To our knowledge, this is the first work to fine-tune an LLM for directly solving diverse CO problems from natural-language descriptions.
>
> **W2. Only OPRO is included as a baseline, despite the existence of other approaches(lines 87-90). Additionally, it is unclear why solvers like Gurobi and LKH are not included**
>
> Thank you for the thoughtful comment. We address both aspects below.
>
> **LLM-based end-to-end solvers:**
>
> We reviewed the relevant literature and cited key works (lines 87–90). Two are benchmarking studies evaluating general-purpose LLMs on small or synthetic CO tasks using zero-/few-shot prompting—aligned with our baselines (e.g., GPT-4o, Claude, DeepSeek), which we evaluate using consistent one-shot prompting for fairness.
>
> A nice recent work, [35], uses “graph thought” reasoning for small TSP instances (10–20 nodes) but reports a 3.6% optimality rate. In contrast, our method solves 96% of TSP instances at this scale within 1% optimality gap, showing significant gains. Additionally, [35] does not release code or models, preventing direct comparison.
>
> OPRO [19] and LMEA [34] represent two distinct strategies: OPRO treats LLMs as verbal optimizers using prompt gradients, while LMEA uses LLMs as evolutionary operators. Both were originally developed for TSP. In our work, we extended OPRO to all 7 CO tasks. To strengthen the comparison, we further adapt LMEA to the same task suite.
>
> In each LMEA generation, the LLM selects parent solutions from the population and performs crossover and mutation via prompting. We designed tailored prompt-based operators to handle the differing solution structures across CO problems. As shown in the table below (reported as fea.(opt.)), our models still significantly outperform LMEA across all tasks.
>
> |Method|        TSP     |       CVRP     |         OP       |       MIS     |       MVC      |        PFSP     |     JSSP      |
> |----------|-----------------|------------------|------------------|---------------|------------------|-------------------|-----------------|
> | LMEA | 77%(265%) |24%(61.24%)|48%(66.18%)|5%(25.0%)|13%(34.22%)|98%(14.31%) | 44%(83.19)|
> | Ours  | 100%(1.07%) |  100%(4.53%)  | 100%(1.85%)   |  94%(1.04%) |  100%(1.29%) | 100%( 1.03%) |  100%(8.20%) |
>
> **Domain-specific solvers:**
>
> We agree that comparing with traditional solvers is important, as also requested by Reviewer egzu. We compare our method with leading domain-specific (meta)heuristics and Gurobi. We evaluate LKH for TSP, HGS for CVRP, EA4OP for OP, and QIG for PFSP.  The results are shown below in the form of opt.(time). Notably, our method achieves better performance than Gurobi on routing problems with a similar time budget.
>
> |                           |       TSP    |    CVRP   |       OP      |       MIS      |    MVC      |   PFSP  |    JSSP   |
> |--------------------|---------------|---------------|---------------|----------------|---------------|--------------|-----------|
> | SOTA method |0.00%(0.3)|-0.19%(1.2)| 1.60%(0.3)|       -          |          -       |0.00%(2.0)| - |
> | Gurobi |1.82%(9.3)| 35.09%(9.8 )|21.96%(9.8)|0.00%(1.0)|0.00%(1.0)|0.46%(9.3)|2.45% (10.0)|
> |         Ours      |1.07%(11.7)|4.53%(11.3)|1.85%(10.8)|1.04%(4.0)|1.29%(7.1)|1.03%(3.1)| 8.20%(20.7)|
>
> However, we emphasize that the goal of our work is not to surpass all the state-of-the-art solvers. While our method may not outperform highly specialized algorithms, it offers a **user-friendly, language-driven interface** that requires **no coding or domain-specific formulations**. Though it may not always match the optimality of LKH or Gurobi, it delivers strong performance with high feasibility, low optimality gaps, and broad generalizability across multiple CO tasks.
>
> **Q1. Should all LLM-based methods enforce equal sampling with the same N? It is better to explicitly report how many samples are generated per instance, whether BoN is used, and if multi-round prompting (e.g., OPRO, PHP) is applied.**
>
> In our experiments, the general-purpose and reasoning LLMs are configured to generate a single solution per instance, aligning with our SFT+RL method, which also produces **one generation without BoN**.  Therefore, the equality holds for the sampling budget of the LLM-based methods. Given the equality, our method achieves significantly better performance in both feasibility and optimality, as shown in Table 1. This indicates that our fine-tuned model not only performs well under equal constraints but also provides robust solutions without relying on test-time sampling tricks.
>
> For the prompting strategies, we have indicated their settings in Appendix D. For example, OPRO runs 4 iterations, generating 5 random solutions each (20 total). They inherently perform multi-round or multi-sample generation, so we do not apply BoN. Despite their extensive exploration, our one-shot SFT+RL model still significantly outperforms them in solution optimality, showing the power of end-to-end learning over handcrafted prompting.
>
> To further address your concern, we also conducted a BoN vs. BoN comparison using DeepSeek-R1, the strongest reasoning model baseline. The results below show that with 8 samples per instance for both models, our method still outperforms DeepSeek-R1:
>
> |           Method       |        TSP       |        OP        |       CVRP       |        MIS       |     MVC      |      PFSP    |       JSSP     |
> |-------------------------|-------------------|-----------------|--------------------|------------------|----------------|---------------|---------------|
> | DeepSeek + BoN |75%(55.43%)|68%(41.73%)| 44%(32.65%) | 62%(1.81%) |51%(4.02%)|  100%(13.31%) | 10%(25.20%)|
> |            Ours          | 100%(1.07%) |  100%(1.85%)  | 100%(4.53%)   |  94%(1.04%) |  100%(1.29%) | 100%( 1.03%) |100%(8.20%) |
>
> **Q2. In Appendix E.7, do the baseline methods leverage any prior information**
>
> There are three types of baselines involved in this section: 1) heuristics (e.g., FIFO and ATC), 2) evolving-based heuristics (e.g., ReEvo and MCTS-AHD), and learning-based methods (e.g., L2D and StarJob).
>
> Among these, the **evolving-based** and **learning-based** methods do leverage prior training data. Specifically, evolving-based methods such as ReEvo and MCTS-AHD use LLMs to generate algorithmic components, which are then evaluated on a training set to compute fitness scores during evolution. This means their algorithmic design is explicitly influenced by performance on prior instances.  Learning-based methods such as L2D and StarJob train neural models or LLMs on large synthetic CO datasets via supervised or reinforcement learning, deeply tying them to training data. It means all the baseline methods are benefiting from prior knowledge of the instances. Despite this, our method still outperforms them all.
>
> **Q3. Should all TSPLIB instances that meet certain criteria be considered**
>
> In our experiments, we select TSPLIB instances with fewer than 100 nodes and EDGE_WEIGHT_TYPE = EUC_2D, i.e., defined by 2D Euclidean coordinates. We exclude instances without this property (e.g., non-Euclidean or distance-matrix-only) because our model is trained on continuous 2D coordinates.
>
> However, training a similar model using **distance matrices** is also possible by adapting input representations or fine-tuning a model variant on such data. We see this as a promising direction for future work, especially for generalized TSP variants and real-world cases lacking coordinate data

---

> > ### Author Response · Authors · 2025-08-01
> > **References**
> >
> > [1] Chen, Qingyu, et al. "Benchmarking large language models for biomedical natural language processing applications and recommendations." Nature Communications 16.1 (2025): 3280.
> >
> > [2] Chen, Nuo, et al. "Graphwiz: An instruction-following language model for graph computational problems." Proceedings of the 30th ACM SIGKDD Conference on Knowledge Discovery and Data Mining. 2024.

---

> > ### Comment · Reviewer_umUz · 2025-08-07
> >
> > Thank you for the detailed clarifications and additional experiments. All of my concerns have been satisfactorily addressed, and I will increase my score accordingly.

---

> > > ### Author Response · Authors · 2025-08-07
> > >
> > > Thank you again for your recognition of the motivation, outstanding performance, and comprehensive experiments. We truly appreciate you raising the score!

---

### Official Review · Reviewer_egzu · 2025-07-03

**Clarity:** 3
**Significance:** 3
**Originality:** 3
**Rating:** 5
**Confidence:** 4

**Summary:**

This paper presents a two-stage fine-tuning framework that turns a 7B-parameter LLM into an end-to-end solver for combinatorial optimization problems. First, SFT teaches solution patterns from solver-generated examples. Second, a feasibility-and-optimality-aware reinforcement learning (FOARL) stage uses a combined constraint-satisfaction and optimality reward to correct infeasible outputs. At test time, Best-of-N sampling further improves solution quality. Empirically, the method achieves near-100% feasibility and optimality gaps of 1–8% across seven CO problems on small to medium sized instances, outperforming both general-purpose, reasoning LLMs and previous LLM as optimizer works.

**Questions:**

1- You compare primarily to NN, FI, sweep, ACO, etc. Why not include modern VRP solvers such as HGS or ALNS, which are widely regarded as state-of-the-art for CVRP? (Similarly for other problems.) Without them, claims of “outperforming problem-specific heuristics” are overstated.

2- Appendix E.1 reports LLM inference times, but you do not compare to MIP solvers like Gurobi/CPLEX on the same tasks. For your instance sizes the exact solutions should be tractable (at least given a target gap@k). Can you provide both solution quality and runtime from a standard solver to justify the benefit of your pipeline?

3- The heuristic features are expert-crafted and problem-specific. How sensitive is your method to the choice or number of such features? How do they affect generalization across problem types if they must be designed by an expert? Could heuristic-feature selection itself be treated as parameter tuning?

4- FOARL improves feasibility, but how does the number or type of constraints affect infeasibility rates? For instance, if you add more constraints (such as time windows or precedence constraints in VRP), does performance degrade gracefully, or not?

5- You evaluate on small/medium/large graphs (Table 2) but, if I understand correctly, always train and test on the same scale. Have you tried training on one size or distribution (e.g., uniform coordinates) and testing on another (e.g., clustered or larger instances) to assess out-of-distribution robustness?

**Ethical Concerns:**

["NO or VERY MINOR ethics concerns only"]

**Final Justification:**

The proposed LLM-as-solver method is not competitive with (1) general-purpose solvers or (2) problem-specific SOTA heuristics, and its scalability is limited (e.g., fails on 1,000-node TSP with huge gaps)

However, the stated goal is not to replace specialized/general solvers for complex, professional-grade problems, but to assist non-experts with simpler, everyday optimization tasks.

Within this scope, the results are significant:

- The method clearly outperforms prior LLM-as-solver approaches. Better gaps and larger problems.
- Rebuttal experiments show it also surpasses “model-then-solve” methods with similar aims.

Overall, this is a clear step forward in LLM-as-solver research and merits publication.

**Limitations:**

Yes.

**Quality:**

3

**Strengths And Weaknesses:**

Strengths
- First to directly map natural-language CO descriptions to solutions via SFT + RL without intermediate code execution.
- Solid integration of SFT and RL with a clear mathematical formulation. Introduces FOARL to explicitly enforce constraints in language outputs.
- Thorough experiments: seven diverse CO tasks, 3 different problem sizes, BoN sampling study, ablations on heuristic features, and runtime breakdown.
- Methodology clearly described; prompt templates, reward definitions, LoRA settings, and problem-instance conversions are fully documented (Sections 4, Appendices A-C).
- Demonstrates that a relatively small LLM can match or exceed much larger LLMs on CO tasks in feasibility and solution quality.


Weakness:
- Claims of outperforming “problem-specific heuristics” rest on comparisons to simple methods (NN, FI, PS, ACO) rather than modern state-of-the-art metaheuristics (e.g., HGS, ALNS). No justification is given for excluding these stronger baselines.
-The description of “heuristic features” (e.g., top-k nearest neighbors) is brief and lacks detail on selection rationale or how different feature choices affect performance and generalization.
- The paper does not benchmark against exact solvers (e.g., Gurobi, CPLEX) on any task, leaving it unclear when one would prefer this approach over off-the-shelf MIP solvers.
- For the problem sizes addressed, the computational efficiency might still be lower than both classical heuristics and MIP solvers, raising doubts about practical value. Further experiments are needed.

---

> ### Author Rebuttal · Authors · 2025-07-30
>
> Dear Reviewer egzu,
>
> Thank you for the valuable comments and for recognizing the solid methodology, thorough experiments, and clear description in our paper. We will address your concerns below.
>
> **[W1.1., Q1] No justification is given for excluding stronger baselines (e.g., HGS, ALNS)**
>
> Thanks for raising the point. We acknowledge the strength of advanced metaheuristics like HGS, which are highly effective for specific problems and often used to generate (near-)optimal reference solutions in previous literature. However, their specialization limits their versatility compared to our approach. Our goal is not to outperform these specialized algorithms in terms of raw solution quality. Some of them, such as LKH for TSP, are actually used in our pipeline to generate high-quality data for training. Rather, we aim to show that LLMs can learn to serve as *general-purpose, end-to-end, language-driven CO solvers*, without requiring problem-specific algorithm design, expert-crafted heuristics, or specialized solver integration.
>
> Since our method does not yet outperform top metaheuristics in optimality, we focused on widely used heuristics that better **reflect the trade-offs between solution quality, generality, and human effort**. Thus, we used “problem-specific heuristics” instead of “problem-specific metaheuristics”. To the best of our knowledge, this already marks a significant step forward for LLMs in addressing CO problems, while prior approaches have not even surpassed simple heuristics [1,2].
>
> However, for your information, we have evaluated some advanced metaheuristics or local search methods. We evaluate LKH for TSP, HGS for CVRP, EA4OP for OP, and QIG for PFSP. Due to the lack of open-source implementations or widely used SOTA heuristics, we use Gurobi for MIS, MVC, and JSSP. The results are summarized below (in the form of opt.(time)):
>
> |                           |       TSP    |    CVRP   |       OP      |       MIS      |    MVC      |   PFSP  |    JSSP   |
> |----------------------|---------------|---------------|---------------|----------------|---------------|--------------|-----------|
> | SOTA method   |0.00%(0.3)|-0.19%(1.2)| 1.60%(0.3)|0.00%(1.0)|0.00%(1.0)|0.00%(2.0)|2.45%(10.0)|
> |         Ours        |1.07%(11.7)|4.53%(11.3)|1.85%(10.8)|1.04%(4.0)|1.29%(7.1)|1.03%(3.1)| 8.20%(20.7)|
>
>
> **W1.2. The description of “heuristic features” is brief and lacks detail on selection rationale**
>
> The heuristic features in our work are deliberately simple, and aligned with domain knowledge to guide LLMs toward promising regions of the solution space, without relying on explicit multi-step reasoning or solver-specific templates. Broadly, they reflect **standard greedy principles** found in traditional heuristics. For example, the features often mirror decisions a classical greedy solver would make at each step. Our selection is guided by two main reasons:
>
> (1) **They are easily computable and widely used in classical heuristics**. For example, top-k nearest neighbors are often used in a well-established greedy strategy for routing, while top-k jobs with the shortest processing times are commonly used in scheduling problems. These features embody canonical greedy schemes that are efficient and effective in many practical scenarios.
>
> (2) **They are naturally compatible with the language modality**. Unlike complex heuristics (e.g., metaheuristics), which are difficult or inefficient to encode in text, our greedy cues can be expressed briefly by text. This ensures they can be included in the LLM input without significant token overhead or representational ambiguity.
>
> **[W2., Q2] The paper does not benchmark against exact solvers**
>
> We agree that comparing against exact solvers is valuable for practitioners to better understand the trade-offs. To this end, we benchmark the CO tasks in our paper using Gurobi. We report results with time limits of 1s and 10s, with the latter roughly matching the average computation time of our method. The results are summarized below:
>
> | Method           |   TSP  |   CVRP |     OP    |    MIS   |   MVC   |  PFSP  |  JSSP  |
> |--------------------|----------|------------|------------|-----------|-----------|-----------|-----------|
> | Gurobi (1s)     | 5.76% |         -    | 49.42% |  0.00% |  0.00% |  2.28% | 41.34%|
> | Gurobi (10s)   | 1.82% | 35.09% | 21.96% |  0.00% |  0.00% |  0.46% |  2.45% |
> | Ours               | 1.07% |  4.53%  | 1.85%   |  1.04% |  1.29% |  1.03% |  8.20% |
>
> As expected, Gurobi performs very well on certain problems, particularly MIS and MVC, where the MIP formulations are tight and effective. However, on more complex problems like routing, our method significantly outperforms Gurobi **under similar time budgets**. Notably, our model achieves nearly **8× smaller optimality gaps** than Gurobi on CVRP (4.53% vs. 35.09%) and solves OP with far better quality.
>
> Importantly, our method offers several advantages: 1) **Ease of use**: there is no need for formal modeling, solver licenses, or programming interfaces. 2) **Language-driven interaction**: users can specify problems in natural language, making it more accessible for non-experts. 3) **Generalizability**: a single model architecture handles diverse problems without customization. While **exact solvers remain preferred in applications demanding guaranteed optimality** with long runtime, our method offers a compelling alternative when flexibility or rapid prototyping is prioritized.
>
> **W3. The computational efficiency might still be lower than both classical heuristics and MIP solvers**
>
> Indeed, constructive heuristics like NN are extremely fast, often producing solutions within 1s. However, as shown in Table 2, our method significantly outperforms them in solution quality. For instance, on large-graph TSP, our model achieves nearly 20× smaller optimality gaps compared to NN. This suggests that the modest trade-off in inference time yields substantial gains in solution quality.
>
> We have also compared our model against Gurobi (see the response to W2), and the results show that, under a similar time budget, our method delivers comparable or even better performance on certain tasks.
>
> It is also important to note that the core contribution of our paper is not superior efficiency, but the introduction of a unified, language-based paradigm for CO. Unlike traditional solvers, our approach requires no problem modeling or coding, and can generalize across problems using a single fine-tuned model.
>
> **Q3. The effect of heuristic features**
>
> Indeed, there is a trade-off between the number of heuristic features and the performance of the model. While incorporating more features may improve solutions, it also increases input length, leading to higher token usage and training cost. In this sense, feature selection can **indeed be treated as a form of parameter tuning**, as you noted.  In Appendix E.4, we presented an ablation analysis to study the performance with and without these features. Taking TSP as an example, we further study the impact of different numbers of heuristic features: We vary the value of k from 0 to 2, and the optimality gaps are [1.17%, 1.10%, 1.07%], while the training time (min) per 1K instances are [75, 112, 150].
>
> **Regarding generalization**: our features represent **simple greedy patterns**, such as NN relationships in routing or shortest processing times in scheduling, which are well-known, easy to compute, and generally applicable. As such, incorporating them does not require much domain expertise.
>
> **Please note: **even** without these features**, our model still outperforms all general-purpose LLMs.
>
> **Q4. FOARL improves feasibility, but how does the number or type of constraints affect infeasibility rates**
>
> This is a very insightful question. Our current study focuses on a set of well-established problems that include common constraint types (e.g., capacity constraints in CVRP).  We emphasize that these NP-hard problems are more difficult than some previous benchmark studies, and this shows an impressive improvement of LLM in solving intractable problems.
>
> Due to the time limitations of the rebuttal phase, it is not feasible to generate new datasets and retrain models for tasks with more and diverse constraints. But we are happy to explore this in future work.
>
> We also recognize that different constraint types, such as arithmetic, logical, or global constraints, may pose varying levels of difficulty for LLM reasoning and RL-based correction. Exploring constraints themselves can be an independent work, which is a bit out of the scope of this work.
>
> From our current results, we observe that FOARL improves feasibility, especially on problems with inequality-type constraints, such as in OP and CVRP. As discussed in Section 5.4, we find that RL plays a role similar to a constraint relaxation mechanism, guiding the model toward feasible regions even when initial generations are suboptimal. We believe the discussion there already offers some insights on the influence of constraints.
>
> **Q5. OOD performance**
>
> We agree that OOD performance is important for a CO solver. To this end, we evaluate the models on routing problems with two OOD distributions, namely cluster and mixed distributions [3]. We set the number of clusters to 7, and the results are summarized below, indicating that our models still perform well when there is a distribution shift.
>
> metric              | tsp-fea.      | tsp-opt.       | op-fea.     | op-opt.       | cvrp-fea.     | cvrp-opt.
> ---------------------|----------|------------|----------|------------|----------|------------
> In-distribution          | 100%     | 1.07%      | 100%     | 1.85%      | 100%     | 4.53%
> OOD (clustered)     | 100%      | 1.41%      | 100%     | 1.89%      | 100%     | 5.12%
> OOD (mixed)          | 100%     | 1.23%      | 100%     | 2.30%      | 100%     | 5.04%
>
> **We will incorporate all the results and discussion above into the paper for clarity.**

---

> > ### Author Response · Authors · 2025-08-01
> > **References**
> >
> > [1] Tang, Jianheng, et al. "GraphArena: Evaluating and Exploring Large Language Models on Graph Computation." The Thirteenth International Conference on Learning Representations.
> >
> > [2] Yang, Chengrun, et al. "Large language models as optimizers." The Twelfth International Conference on Learning Representations. 2023.
> >
> > [3] Li, Sirui, Zhongxia Yan, and Cathy Wu. "Learning to delegate for large-scale vehicle routing." Advances in Neural Information Processing Systems 34 (2021): 26198-26211.

---

> > ### Comment · Reviewer_egzu · 2025-08-05
> >
> > I appreciate your detailed reply and the effort you put into conducting the additional experiments.
> >
> > **[W1.1, Q1]:** There is a blurred line between heuristics and metaheuristics. For example, why do you consider LKH a metaheuristic when its framework is not generalizable to other CO problems? I still believe it is unfair to claim that your method outperforms problem-specific heuristics.
> >
> > **\[W2, Q2]:** Benchmark against exact solvers. Please include these results in the paper along with the mathematical models used. As you know, for the VRP there are several formulations, and the choice of formulation is critical. Please see [1] as an alternative VRP formulation, which is known for its fast convergence on modern solvers.
> >
> > **Q3: The effect of heuristic features.** Using these features, even simple greedy rules (e.g., top-k nearest neighbors in routing or top-k jobs with the shortest processing times), makes your methods less general and problem-independent than you claim. A user still needs substantial familiarity with the specific CO problem to identify which heuristic features are common and impactful. These features are coming from the common pattern in the optimal solution of the CO problem.
> >
> > In general, there is no justification for using LLMs to solve combinatorial optimization problems, especially since their performance is not comparable with general-purpose solvers in terms of both feasibility and optimality gap, and the user still requires background knowledge of the CO problem. However, I am willing to give your work a chance, since this is a new direction and future research may achieve superior performance.
> >
> > [1] Baldacci, R., Hadjiconstantinou, E., & Mingozzi, A. (2004). An exact algorithm for the capacitated vehicle routing problem based on a two-commodity network flow formulation. *Operations Research, 52*(5), 723–738.

---

> > > ### Author Response · Authors · 2025-08-06
> > > **Thank you for your follow-up**
> > >
> > > Dear Reviewer egzu,
> > >
> > > Thanks for your thoughtful follow-up. We appreciate your continued engagement and constructive suggestions, and we are happy to provide further clarification on the points you raised.
> > >
> > > **[W1.1, Q1: On LKH and fairness of heuristic comparisons]**
> > >
> > > In fact, LKH3 relies on many operators, i.e., different $k$ levels of $k$-opt, with different strategies (e.g., nested loop that can be regarded as variable neighborhood search). To some extent, it is towards metaheuristics rather than simple local search (such as the simple 2-opt operator). From this perspective, it is a type of metaheuristic that can be applied to broad VRPs, keeping generalized to more routing problems (see its website).  We included it in our comparisons due to its exceptional performance and widespread use as a benchmark for TSP.
> > >
> > > However,  following your suggestion, in our rebuttal, we referred more broadly to “local search and metaheuristic” methods, and we agree that this terminology should be used more precisely.
> > >
> > > Regarding fairness: we fully acknowledge that our method is not intended to outperform the best-in-class, problem-specific solvers like LKH or HGS. These algorithms have been refined over decades for their target domains. Our claim can be more modest: we will revise the phrasing in the paper to state that **our model outperforms the simple heuristics listed in Table 2**, rather than suggesting it surpasses problem-specific heuristics. We hope this clearer and more accurate statement better reflects the scope of our contribution.
> > >
> > >
> > > **[W2, Q2: Benchmark against exact solvers and model clarity]**
> > >
> > > Thank you for highlighting this important point. We agree that exact solvers are a critical baseline, and that the choice of mathematical formulation for problems like VRP significantly impacts solver performance.
> > > We currently adopt a **three-index vehicle formulation** (also referred to as the vehicle-indexed formulation) with MTZ subtour elimination constraints. We will update the final version of the paper to include a clear description of the mathematical models used for each problem and incorporate results from exact solvers (including runtime and optimality gap where applicable). We also appreciate the reference you provided and will explore benchmarking using the two-commodity flow formulation.
> > >
> > > **[Q3: On the use and generality of heuristic features]**
> > >
> > > We incorporated these features for better performance, but it does not mean that the features are necessary for obtaining a satisfactory solution. To evaluate the model’s robustness, we conducted ablations (Appendix E.4), which show that **removing these features results in only a marginal drop in performance**—for example, just a 0.1% increase in optimality gap on TSP. Thus, while such features help, they are **not essential** for our model to outperform existing LLM-based approaches. We believe this demonstrates the method’s resilience and accessibility, even without task-specific tuning.
> > >
> > > **[Final Note: On the justification for LLMs in CO]**
> > >
> > > We sincerely appreciate your openness in acknowledging this as a new direction. Our goal is not to position LLMs as replacements for expert-designed solvers but to explore their potential as **general-purpose, language-driven optimizers**, particularly for users who may lack programming skills or solver knowledge. While current performance does not yet match SOTA solvers, we believe the usability, interactivity, and accessibility of our approach will make it increasingly relevant as LLMs evolve
> > >
> > > In fact, LLMs in CO have been studied a lot [1], which are motivated by their ability to automate CO problem solving [1, 2]. Our work is a natural further step from the previous work and aims to develop a language-based end-to-end optimizer. While it is not better than domain-specific solvers (no current LLM-based work has such performance), we see a significant advantage of our work over existing LLM-based work. We believe this new research direction would encourage more work in the future to push the performance frontier of LLMs in CO.
> > >
> > > Thank you again for your feedback. We will revise the paper accordingly and are open to clarifying any of your further concerns. Please also take our response into account.
> > >
> > > *[1] Da Ros, Francesca, et al. "Large Language Models for Combinatorial Optimization: A Systematic Review." arXiv preprint arXiv:2507.03637 (2025).*
> > >
> > > *[2] Liu, Fei, et al. "Evolution of Heuristics: Towards Efficient Automatic Algorithm Design Using Large Language Model." International Conference on Machine Learning (2024).*

---

> ### Author Response · Authors · 2025-08-09
> **Your feedback would be appreciated**
>
> Dear Reviewer egzu,
>
> As the discussion phase is set to close in less than 8 hours, we want to ensure that all your questions and concerns have been fully addressed. We have carefully reviewed your feedback and provided detailed responses to each of your comments. We sincerely hope our clarifications meet your expectations and resolve any outstanding issues.
>
> If you have any further concerns, we will also do our best to address them. Thank you for your valuable feedback. We look forward to hearing from you!
>
> Sincerely,
>
> Authors of Submission15371

---

### Note · Authors · 2025-08-15

We sincerely thank all reviewers for their insightful comments. We also appreciate their recognition of the strengths of our proposed framework, including its novelty, comprehensive experiment, strong performance, and clear writing.

During rebuttal, we addressed all comments with detailed explanations and extensive results. However, we didn’t receive feedback on our final responses, leaving them unacknowledged. To help recap the rebuttal, we summarized the last two questions that we have addressed:

**Reviewer qC4e required a comparison with "model-then-solve" pipelines.**

1.	Our comprehensive experiments have been acknowledged by reviewers, which are more than enough to support the scientific focus of this paper, that is, developing an end-to-end LLM CO optimizer without solver invocation.

2.	Following the literature such as [1,2], we didn’t compare with "model-then-solve" pipelines due to their immaturity, including modeling errors and inferior performance. Please refer to our response titled “Official Comment by Authors” on Aug 6.

3.	Despite the unnecessariness to compare, we follow the reviewer’s suggestion to extensively compare with recent SOTA “model-then-solve” methods, demonstrating that our approach can achieve better performance (see our response “More results to verify our method outperforms ‘model and solve’ SOTA”). We will include these results in our final version.

**Reviewer u6T1 required a comparison with supervised GNNs.**

1.	We emphasize that our work targets training the first end-to-end LLM solver for CO, so we focused our comparison with LLM-based methods and domain-specific CO solvers.

2.	Following the reviewer’s suggestion, we compared with supervised GCN [3], which is a GNN specifically for TSP (see our response “Official Comment by Authors” on Aug 6). We also compared with GREAT, which is a recent SOTA but is still on arXiv. Again, the comparison showed the strong performance of our method.

We believe these detailed responses can address the remaining concerns. We thank the reviewers again for their time, thoughtful engagement, and constructive input.


[1] Tang, Jianheng, et al. "GraphArena: Evaluating and Exploring Large Language Models on Graph Computation." ICLR. 2025

[2] Iklassov, Zangir, et al. "Self-guiding exploration for combinatorial problems." Neurips. 2024

[3] Joshi, Chaitanya K., et al. "An efficient graph convolutional network technique for the travelling salesman problem." arXiv:1906.01227.

---

### Decision · Program_Chairs · 2025-09-17

**Decision:**

Accept (poster)

**Comment:**

The paper proposes a two-stage fine-tuning framework that enables Large Language Models (LLMs) to serve as end-to-end combinatorial optimization (CO) solvers. The method utilizes supervised fine-tuning (SFT) and a feasibility-and-optimality-aware reinforcement learning (FOARL) process to directly map natural language problem descriptions to solutions, eliminating the need for external solvers or code generation.
Most reviewers acknowledged the framework's novelty, clear methodology, and comprehensive experiments. Although Reviewer qC4e did not provide further feedback during the discussion phase, their final concern regarding a lack of convincing evidence against the "model-then-solve" approach was addressed by the authors. As noted by Reviewer egzu, the authors conducted additional experiments compared to several "model-then-solve" baselines (e.g., LLMOPT, ORLM, and DRoC), demonstrating their method's superior performance. Overall, the authors have addressed all major concerns raised by the reviewers. Therefore, I recommend accepting this paper.